# LEARNING-TO-MEASURE: IN-CONTEXT ACTIVE FEATURE ACQUISITION

## ABSTRACT

Active feature acquisition (AFA) is a sequential decision-making problem where the goal is to improve model performance for test instances by adaptively selecting which features to acquire. In practice, AFA methods often learn from retrospective data with systematic missingness in the features and limited task-specific labels. Most prior work addresses acquisition for a single predetermined task, limiting scalability. To address this limitation, we formalize the meta-AFA problem, where the goal is to learn acquisition policies across various tasks. We introduce Learning-to-Measure (L2M), which consists of i) reliable uncertainty quantification over unseen tasks, and ii) an uncertainty-guided greedy feature acquisition agent that maximizes conditional mutual information. We demonstrate a sequence-modeling or autoregressive pre-training approach that underpins reliable uncertainty quantification for tasks with arbitrary missingness. L2M operates directly on datasets with retrospective missingness and performs the meta-AFA task in-context, eliminating per-task retraining. Across synthetic and real-world tabular benchmarks, L2M matches or surpasses task-specific baselines, particularly under scarce labels and high missingness.

## 1 INTRODUCTION

Machine learning (ML) methods typically operate under the assumption that all input features are available at inference time. However, this assumption does not hold in scenarios where acquiring certain features involves significant costs or risks, such as medical diagnostics (Erion et al., 2022). For example, acquiring imaging data or invasive biopsies may incur substantial financial costs and pose potential risks to patient safety (Callender et al., 2021). In such cases, there is a need to adaptively determine the value of feature acquisition against its costs to make informed decisions.

Active feature acquisition (AFA) addresses this problem by learning an agent to adaptively select which features to acquire or observe for each sample (Ma et al., 2018; Shim et al., 2018; von Kleist et al., 2023b). AFA is naturally a sequential decision-making problem, where conditioning on past feature acquisitions can inform collection in the future. Prior AFA work uses either greedy acquisition strategies that maximize an estimate of the one-step expected information gain (Ma et al., 2018; Gong et al., 2019; Covert et al., 2023; Chattopadhyay et al., 2023; Gadgil et al., 2023), or RL approaches that learn a value (or Q-) function to improve multi-step feature acquisition (Shim et al., 2018; Kachuee et al., 2019; Janisch et al., 2019; Li & Oliva, 2021).

Most AFA methods suffer from common bottlenecks. First, they are trained on retrospective data, which consists of intrinsically missing features (von Kleist et al., 2023a;b). For example, clinical data are often incomplete and shaped by clinical protocols, resource constraints, workflow decisions, and patient behavior. This leads to systematic missingness in features across subpopulations and limited task-specific data labels (Jeanselme et al., 2022; Chang et al., 2024; Zink et al., 2024). Consider chest-pain triage in the emergency department, where the guidelines typically prioritize first-line laboratory tests, followed by additional invasive testing or X-ray studies, leading to missingness dependent on past observations. Agents trained on retrospective data may encode the same acquisition bias. Imputing features will not calibrate on the informativeness of the feature; likewise, models that ignore missingness without modulating uncertainty will replicate missingness patterns.

Figure 1: Schematic of the meta-AFA problem. The model can acquire lab measurements for multiple tasks, Tasks 1-3. The input is a sequence of past observations together with a query, and the policy $\pi_\theta$ outputs the next greedy acquisition action. The updated query is then used for the following step. After $k$ acquisitions, the predictor $f$ produces the final label prediction.

Second, existing AFA methods target single predetermined tasks rather than a general capability aligned with the foundation model paradigm (Bommasani et al., 2021). Prior approaches also rely on complex latent-variable models with heuristic approximations to make generative modeling feasible. These methods typically obtain uncertainty through posterior sampling, which is often unreliable, especially in high-dimensional settings (Ma et al., 2018; Li & Oliva, 2021; Peis et al., 2022).

To address these challenges, we introduce Learning-to-Measure (**L2M**), a new in-context AFA approach building on the uncertainty quantification capabilities of pre-trained sequence models (Nguyen & Grover, 2022; Ye & Namkoong, 2024; Mittal et al., 2025). At its core, **L2M** couples uncertainty quantification with a greedy decision policy for selecting the next acquisition action. **L2M** operates directly on datasets with retrospective missingness and solves the AFA task in-context, provided the missingness satisfies certain assumptions such as missing at random (MAR) and the data contains sufficient coverage of acquisitions.

**L2M** consists of two stages: (i) pretraining across tasks with missingness to quantify predictive uncertainty of a target variable given partially observed inputs, and (ii) meta-training a policy network to greedily acquire features that reduce the predictive uncertainty via a smooth, differentiable approximation to information gain, enabling end-to-end optimization. We implement the first state using sequence modeling over data sequences to capture reliable beliefs under missingness. This design yields a principled approach to sequential information acquisition across tasks. **L2M** removes latent-variable approximations and performs calibrated, scalable uncertainty estimation via direct sequence prediction. Figure 1 depicts the schematic of the **L2M** framework at inference.

Our contributions are the following:

1. **Meta-learning AFA across diverse tasks and missingness patterns:** We formalize the problem of meta-learning AFA policies across (time-invariant) tasks with diverse input data distributions and retrospective missingness mechanisms.

2. **Combining uncertainty estimation and decision-making via sequence modeling:** We propose **L2M**, a scalable transformer-based approach for end-to-end sequential information maximization. The sequence model provides reliable uncertainty estimates for partially observed inputs and leverages these estimates to predict the next optimal feature to acquire. To learn the policy, we design a smooth, differentiable approximation of the acquisition problem, resulting in a fully auto-differentiable training framework.

3. **Robustness to limited labeled data and missingness:** We empirically show that our meta-learning-based approach, **L2M**, outperforms task-specific baselines across tasks of varying sizes and degrees of missingness, particularly when labeled data is scarce and feature missingness is high.

In the following, we first introduce the meta-AFA problem setup in Section 3. Next, we extend meta-AFA to settings where tasks can contain missingness and provide identifiability conditions that allow the optimization problem to be solved using observational data (Section 3.1). Section 4 presents the main components of our proposed solution. We introduce our meta-learning framework using a sequence modeling approach in Section 4.1. Section 4.2 outlines our proposed loss function and the model training procedure.Section 5 demonstrates the empirical utility of our method.

## 2 RELATED WORK

**Active Feature Acquisition (AFA).** Time-invariant AFA methods fall into two main classes:

*Greedy AFA policies:* These methods iteratively acquire features by greedily maximizing the expected information gain (Ma et al., 2018; Covert et al., 2023; Chattopadhyay et al., 2023). For example, Ma et al. (2018); Gong et al. (2019); Chattopadhyay et al. (2022) use generative models to impute potential outcomes of all possible acquisitions and select the greedy action. Covert et al. (2023) uses a policy network to directly predict the greedy action, guided by the loss of a separate prediction model. Gadgil et al. (2023) learns a value network to estimate the information gain directly. Theoretical work have shown that greedy policies achieve near-optimal performance compared to non-myopic ones under certain conditions (Golovin & Krause, 2011; Chen et al., 2015).

*MDP-based policies.* An alternative view treats AFA as a sequential decision-making problem addressed using reinforcement learning (RL). Model-based approaches learn a generative transition model using synthetic rollouts for data-efficient policy learning (Zannone et al., 2019; Li & Oliva, 2021). Model-free approaches directly learn value or Q-functions from offline data, selecting features that maximize expected returns (Shim et al., 2018; Kachuee et al., 2019; Janisch et al., 2019). MDP-based approached are prone to model misspecification, given the challenges of offline value approximation and credit assignment over long acquisition trajectories (Erion et al., 2022).

**AFA under retrospective missingness.** Few studies examine how missingness mechanisms in retrospective data affect feature acquisitions (Ma & Zhang, 2021; von Kleist et al., 2023b). Most prior work either assumes fully observed data or uses simple imputation strategies such as conditional mean imputation, inducing statistical bias in policy evaluation (von Kleist et al., 2023b). Model-based approaches provide a principled alternative when missingness assumptions hold, but face practical limitations: task-specific generative models are hard to estimate with limited data, particularly in high dimensions (Zannone et al., 2019; Li & Oliva, 2021; 2024).

**Meta-learning via Sequence modeling.** Our proposed solution leverages sequence modeling, by framing feature acquisition as an in-context decision-making process rather than relying on explicit generative modeling assumptions. There is a growing body of work formalizing the connection between using sequence models for meta-learning and in-context learning (ICL), and Bayesian inference (Müller et al., 2021; Nguyen & Grover, 2022; Ye & Namkoong, 2024). Other work has applied this framework to decision-making problems (Lee et al., 2023; Lin et al., 2023; Tianhui Cai et al., 2024). The AFA problem differs in that we do not directly observe the reward-maximizing action and must learn policies strictly from offline data.

## 3 THE META-ACTIVE FEATURE ACQUISITION PROBLEM

We consider a supervised learning task where $X \in \mathbb{R}^d$ is a $d$-dimensional feature vector and $Y \in \mathbb{R}$ is the target variable. We assume the features are time-invariant, i.e., the values of $X$ do not evolve over time. We let $X_j$ denote the value of feature $j$ and $X_0$ denote the baseline features that are always observed. At each acquisition step $t \in \{1, \ldots, T\}$, the agent selects an action $A_t \in \{1, \ldots, d\}$, indicating the index of the next feature to acquire. We write $\underline{X}_t = \{X_0, \ldots, X_{A_t}\}$ for the set of acquired features up to and including step $t$. Throughout this work, realizations of random variables are written in lowercase.

*Meta-active feature acquisition (meta-AFA)* trains an agent to acquire features sequentially with the goal of efficiently reducing prediction error for a given task distribution. We assume that each task $\mathcal{T}$ is drawn from an unknown distribution $p(\mathcal{T})$. For each task $\mathcal{T}$, data pairs $(X, Y) \sim p_\mathcal{T}(X, Y)$ are sampled from the task-specific distribution. Given a new task, meta-AFA algorithms seek to sequentially select, at each step $t$, the next feature to acquire based on the partially available information

$\underline{X}_t$, to efficiently reduce the predictive uncertainty on the target variable $Y$. We focus on settings with a fixed budget $k < d$ and (without loss of generality), uniform feature costs.

One common approach to (task-specific) AFA is to acquire features *greedily* based on the *expected* reduction in uncertainty, an approach rooted in Bayesian experimental design (Bernardo, 1979). At each step $t$, the method acquires the feature that maximizes the conditional mutual information (CMI) with the target:

$$I_{\mathcal{T}}(Y; X_j \mid \underline{X}_t = \underline{x}_t) \triangleq \mathbb{E}\left[D_{\mathrm{KL}}\big(p_{\mathcal{T}}(Y \mid X_j \cup \underline{X}_t) \,\big\|\, p_{\mathcal{T}}(Y \mid \underline{X}_t)\big) \mid \underline{X}_t = \underline{x}_t\right]. \tag{1}$$

This requires modeling one-step conditional probabilities $p_{\mathcal{T}}(X_j|\underline{X}_t)$, $p_{\mathcal{T}}(Y|\underline{X}_t, X_j)$ and $p_{\mathcal{T}}(Y|\underline{X}_t)$. In practice, the historical data often contain incomplete observations, complicating the estimation of these conditionals.

## 3.1 TASKS WITH RETROSPECTIVE MISSINGNESS

We refer to inherent missingness in the historical data as 'retrospective missingness', where missingness mechanisms and rates vary across tasks, which can impact AFA performance. We extend the meta-AFA problem to settings with retrospective missingness. Essentially, CMI is only identifiable and can be estimated from data retrospective missingness under certain conditions. We outline these conditions using causal identifiability (Rubin, 1976), which allows us to formalize the conditions under which relevant underlying distributions (to estimate CMI) can be estimated from data with retrospective missingness. We define $R \in \{0,1\}^d$ as the binary missingness indicators corresponding to each feature (Nabi et al., 2020). $X(1)$ is the "potential outcome" of $X$, had $R = 1$ been true, i.e., the measurement had been observed. For a given $X_j(1) \in X(1)$ and corresponding $R_j \in R$, each variable is set by the following deterministic feature revelation mechanism:

$$X_j = \begin{cases} X_j(1) & \text{if } R_j = 1 \\ \text{``?''} & \text{if } R_j = 0 \end{cases}$$

The CMI estimand (Equation 1) can be equivalently denoted as:

$$I_{\mathcal{T}}(Y; X_j \mid \underline{X}_t = \underline{x}_t) \equiv I_{\mathcal{T}}(Y; X_j(1)|\underline{X}_t = \underline{x}_t), \tag{2}$$

and needs to be estimated under $p_{\mathcal{T}}(X(1), Y)$, which is the *reference distribution*, i.e., the joint distribution in the absence of the missingness. The identification of $p_{\mathcal{T}}(X(1), Y)$ from data (with missingness) depends on the missingness mechanism, given by the following assumptions:

**Assumption 3.1.** (*Missing at Random or MAR*) $R_j \perp\!\!\!\perp X_j(1) \mid \underline{X}_t$.

**Assumption 3.2.** (*Exclusion Restriction*) $R_j \perp\!\!\!\perp Y \mid X_j(1), \underline{X}_t$.

**Assumption 3.3.** (*Positivity*) $p(R_j = 1 \mid \underline{X}_t = \underline{x}_t) > 0$ for all values $\underline{x}_t$ and $j \in \{1, ..., d\}$

Intuitively, Assumption 3.1 posits that any systematic differences between observed and missing data can be fully explained by the observed features, rather than by unobserved confounders. Assumption 3.2 states that measuring a feature does not directly affect the target variable. Assumption 3.3 requires sufficient data coverage of each feature acquisition action. These assumptions are analogous to standard assumptions in off-policy evaluation, and yield the following identification result.

**Theorem 3.4.** (*Identification of CMI with retrospective missingness*) *The CMI for any subset $\underline{X}_t \subseteq X$ given by $I_{\mathcal{T}}(Y; X_j(1)|\underline{X}_t = \underline{x}_t)$ is identified when $p_{\mathcal{T}}(X(1), Y)$ is identified. Under Assumption 3.1 (MAR), 3.2 (exclusion restriction), and 3.3 (positivity), the CMI can be estimated by*

$$I_{\mathcal{T}}(Y; X_j(1)|\underline{X}_t = \underline{x}_t) = I_{\mathcal{T}}(Y; X_j|\underline{X}_t = \underline{x}_t, R_j = 1) \tag{3}$$

The proof is provided in Section A.1. Intuitively, if the joint $p_{\mathcal{T}}(X(1), Y)$ is identified, then any functional of the joint is identified. However, estimating these functionals *directly from complete cases* ($R_j = 1$) at step $t$ is valid only under the posited assumptions. The restrictive nature is due to targeting pointwise identification of the greedy action for every $\underline{x}_t$. In practice, this level of generality is often unnecessary as some states are never encountered.

## 4 METHOD

We now present our end-to-end AFA framework (**L2M**) based on amortized optimization and meta-learning a greedy policy with a sequence model parameterized using transformers (Vaswani et al., 2017). We note that our framework can leverage transformers trained from scratch, or pretrained large language models (LLMs). We begin by introducing our proposed Bayesian analog of the CMI objective using sequence models in Theorem 3.4. However, the CMI objective is generally intractable to compute directly. To overcome this, we formulate a tractable surrogate optimization problem that approximates the CMI objective. We then relax the discrete action-selection problem with a smooth, differentiable approximation, which allows us to directly learn the policy using gradient-based methods.

### 4.1 META-LEARNING VIA SEQUENCE MODELING

Formally, in meta-AFA, we consider the set of all test-time query samples with partially observed features, $\{\underline{X}_t^{(q)}\}_{q=m+1}^N$. For each query instance $q$, our goal is to select the next acquisition action based on their currently observed features $\underline{X}_t^{(q)}$ and task-specific context of historical samples $\mathcal{D}_\mathcal{T} = \{X^{1:m}, R^{1:m}, Y^{1:m}\}$. The key challenge of meta-AFA is to model the joint predictive distribution,

$$p(\text{Outcomes} \mid \text{Partial Observations}, \text{Historical Data}) \equiv p(Y^{m+1:N} \mid \underline{X}_t^{m+1:N}, \mathcal{D}_\mathcal{T}),$$

with sufficient flexibility while providing principled uncertainty estimates to guide feature acquisition. Sequence modeling offers a compelling solution: instead of explicitly modeling latent variables, autoregressive training over *data sequences*, together with invariance-inducing inductive biases (Definitions A.8, A.9), provides a practical way to approximate posterior inference from observations alone, building on prior works that have formalized this connection (Nguyen & Grover, 2022; Ye & Namkoong, 2024; Mittal et al., 2025). We note that the sequence model can be meta-learned using synthetically generated tasks (Müller et al., 2021), or using real-world datasets (Gardner et al., 2024).

Sequence modeling decomposes the joint predictive prediction over query samples into a product of one-step conditional probabilities:

$$p(Y^{m+1:N} \mid \underline{X}_t^{m+1:N}, \mathcal{D}_\mathcal{T}) = \prod_{q=m+1}^N p(Y^{(q)} \mid \underline{X}_t^{(q)}, Z^{1:m})$$

where we denote the context for each query sample as $Z^{1:m} = \{X^{1:m}, R^{1:m}, Y^{1:m}\}$, and assume conditional independence across queries given context $Z^{1:m}$ and inputs $\underline{X}_t^{m+1:N}$. Intuitively, by conditioning on a variable-length context containing historical data, the sequence model infers the task-specific mechanism from context and amortizes uncertainty estimation across partially observed queries. Once uncertainty is exactly recovered via the one-step conditional probabilities, the ideal greedy strategy is to acquire the feature with the maximum CMI given by

$$I_\mathcal{T}(Y^{(q)}; X_j^{(q)} \mid \underline{X}_t^{(q)} = \underline{x}_t, Z^{1:m} = z^{1:m}, R_j^{(q)} = 1) \tag{4}$$

Directly maximizing this CMI is impractical because it requires access to the true step-wise conditionals, and expectations over all candidate feature $X_j^{(q)}$. In the following section, we detail our methodology for constructing a surrogate optimization problem using learned approximations.

### 4.2 POLICY OPTIMIZATION

Rather than computing the CMI exactly, we adopt the practical approximation of Covert et al. (2023, Prop. 2): optimize the *one-step-ahead* predictive loss achieved by a predictor $f_\phi$ after acquiring a candidate feature $X_j$. We train a policy $\pi_\theta$ to directly minimize this one-step loss, providing a tractable surrogate for the CMI objective.

To facilitate gradient-based optimization, we consider stochastic policies that output a categorical distribution over actions $\pi_\theta(\cdot \mid \underline{X}_t^{(q)}, Z^{1:m}) \in \Delta^{d-1}$. Additionally, we restrict the learned feature

acquisition policy to "blocked policies", ensuring that features unavailable during training are not sampled by the policy. This removes the need for full generative modeling of the joint $p(X(1), Y)$ to sample missing potential outcomes (von Kleist et al., 2023b):

**Definition 4.1** (Blocked Policy). A blocked policy $\tilde{\pi}_\theta$ is a stochastic policy over feature acquisitions that satisfies the following condition: at each step $t$, it assigns non-zero probability only to features that have support in retrospectively observed data, i.e.,

$$\pi_\theta(j \mid \cdot) = 0 \quad \text{if} \quad R_j = 0,$$

where $R_j = 1$ indicates that feature $j$ is available.

Having restricted attention to blocked policies, we now define the surrogate objective used to jointly train the policy and predictor. Denote the state for the $m$-th sample in the sequence as $S_t^m = (\underline{X}_t^m, \underline{A}_{t-1}^m)$, and $A_t$ is an action sampled from $\tilde{\pi}_\theta(S_t^m, Z^{1:m})$. Let the per-action expected loss be

$$J(a_t; \underline{x}_t^{m+1}, z^{1:m}) = \mathbb{E}_{\substack{X_{a_t}^{m+1}, Y^{m+1} \mid \underline{x}_t^{m+1}, \\ z^{1:m}, R_{a_t}=1}} \Big[ \ell\big( f_\phi(\cdot \mid \underline{x}_t^{m+1} \cup X_{a_t}^{m+1}, z^{1:m}), Y^{m+1}\big) \Big],$$

where we use $\ell$ to denote the log loss (negative log likelihood) for evaluating the predictor. Concretely, the sequence prediction loss is defined as

$$\mathcal{L}(f_\phi, \pi_\theta) = \sum_{\mathcal{T} \sim p(\mathcal{T})} \sum_{m=1}^{N-1} \mathbb{E}_{S_t^{m+1}, Z^{1:m}, R} \Big[ \mathbb{E}_{A_t \sim \tilde{\pi}(\cdot \mid S_t^{m+1}, Z^{1:m})} [J(A_t; \underline{X}_t^{m+1}, Z^{1:m})] \Big]. \tag{5}$$

Our main theorem shows that minimizing the above sequence loss recovers the greedy CMI actions.

**Theorem 4.2** (Surrogate optimality for greedy CMI with context). *Consider the sequence modeling objective in Eq. 5 with cross-entropy loss. Let $Z^{1:m} = (X^{1:m}, R^{1:m}, Y^{1:m})$ be the task-specific context and $\underline{X}_t^{m+1:N}$ the partially observed features for the $N - m$ query points at step $t$. Then any joint minimizer $(\theta^\star, \phi^\star)$ of $\mathcal{L}$ satisfies:*

1. ***Per-query Bayes-optimality.*** *For each $q \in \{m+1, \ldots, N\}$,*
$$f_{\phi^\star}\big(\cdot \mid \underline{X}_t^{(q)}, Z^{1:m}\big) = p\big(Y^{(q)} \mid \underline{X}_t^{(q)}, Z^{1:m}\big).$$

2. ***Step-wise CMI-optimal acquisition for every query.*** *For each query index $q \in \{m+1, \ldots, N\}$ and for $(\underline{x}_t^{(q)}, z^{1:m})$, the policy places mass only on actions that maximize CMI:*
$$j \in \arg \max_{a_t : R_{a_t} = 1} I\Big( Y^{(q)}; X_{a_t}^{(q)} \Big| \underline{X}_t^{(q)} = \underline{x}_t^{(q)}, R_{a_t}^{(q)} = 1, Z^{1:m} = z^{1:m} \Big).$$

*If the maximizer is unique, $\pi_{\theta^\star}(\cdot \mid \underline{x}_t^{(q)}, z^{1:m})$ is a point mass on that action.*

The proof is an extension of the result in (Covert et al., 2023) and is provided in Appendix A.3.

Direct optimization of Equation 5 is non-differentiable because $A_t$ is sampled from a categorical distribution. To obtain gradients, we use a Gumbel–Softmax relaxation of the discrete index sampling operation $a_t \sim \tilde{\pi}_\theta$, which reduces variance of the gradient estimate by introducing bias (Maddison et al., 2016) (compared to REINFORCE (Fu, 2006; Williams, 1992)). We denote the sampled index $A_t \sim \tilde{\pi}_\theta(\cdot \mid S_t^{m+1}, Z^{1:m})$ as $\tilde{A}_t = g_\theta(\eta; S_t^{m+1}, Z^{1:m})$ and reparameterize Equation 5 as follows:

$$\nabla_\theta \mathcal{L}(\theta, \phi) = \mathbb{E}_{S_t^{m+1}, Z^{1:m}, R} \Big[ \mathbb{E}_{\eta \sim \text{Gumbel}(0,1)} J(\tilde{A}_t; \underline{X}_t^{m+1}, Z^{1:m}) \Big], \tag{6}$$

where $\eta \sim \text{Gumbel}(0, 1)$ is the Gumbel distribution and $g_\theta(\eta) = \text{softmax}(\frac{\log \pi_\theta + \eta}{\tau})$. The softmax computation smoothly approaches the discrete argmax computation as $\tau \to 0$ while preserving the relative order of the Gumbels, $\log \pi_\theta + \eta$.

While our method is agnostic to the specific parameterization of sequence models, we use a modified Transformer model with a separate predictor and policy head. Relative to a standard Transformer, we (i) modify the input representation by concatenating masked input features and the missingness indicator mask (which is a common strategy in past AFA work (Covert et al., 2023; Gadgil et al., 2023; Norcliffe et al., 2025) to the label (or padded zeros), (ii) remove positional embeddings, and (iii) replace causal masking with an alternative attention masking structure during both training and inference. The details of our architecture are given in Appendix A.5.

**Training** In practice, our pretraining consists of two stages. Since the optimal predictor $f_\phi$ is independent of the policy $\pi_\theta$, we first pretrain $f_\phi$ on random feature subsets $X_o$ for any $o \subset [d]$, then jointly training the model and policy head. At each training step, we sample a batch of tasks from $p(\mathcal{T})$ and treat them as randomly permuted sequences. Blocking ensures that missing features (with $R_j = 0$) are not acquired by the policy during training. Details of the training procedure are provided in Algorithm 1 and Algorithm 2 (Appendix A.5). Figure 2 (left) summarizes the procedure.

**Inference** After pretraining, the sequence model $\pi_\theta$ is deployed for online AFA decision-making. Algorithm 3 (Appendix A.5) summarizes the feature-acquisition procedure for unseen query samples in a given task, which requires no gradient updates (see Figure 2 (right)).

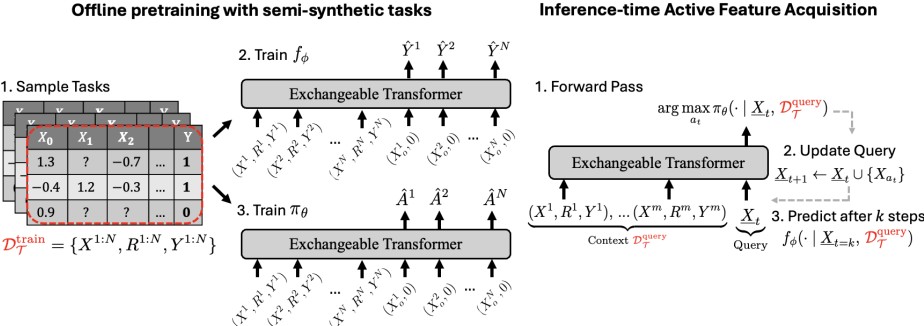

Figure 2: (On the left) Pretraining procedure for the predictor $f_\phi$ and policy $\pi_\theta$. The modified transformer encoder model takes as input task-specific context (a training dataset with retrospective feature missingness) and partially observed queries $X_o^{(j)}$ and predicts the outcomes $Y^{(j)}$ and the optimal greedy action $A^{(j)}$ for the queries. The parameters are learned end-to-end by backpropagating the autoregressive sequence loss over queries. (On the right) The trained transformer model is able to predict labels and optimal actions for unseen query tasks in-context. The attention mask to enable handling different context lengths used in the Transformer is given in Appendix A.5.

# 5 EXPERIMENTS

**Datasets** We aim to demonstrate the feasibility of our **L2M** framework across multiple tasks and diverse applications. While we provide comprehensive experimental details in the Appendix A.6, we provide a brief overview in this section.

First, we train and evaluate on fully synthetic regression tasks sampled from a Gaussian Process (**GP**) prior. Each task is sampled from GPs with randomized RBF kernels $\mathcal{T}_i \sim \mathcal{GP}(m, k)$ with $m(x) = 0$ and contains $d = 10$ features for acquisition. To simulate incomplete observations, features are randomly dropped according to a missing completely at random (MCAR) mechanism. We sample evaluation tasks of varying context lengths from GPs with RBF kernels, as well as Matern kernels, which are unseen during training (Matern).

Next, we evaluate **L2M** on realistic tasks derived from real-world tabular datasets: **Metabric** (Curtis et al., 2012), **MiniBooNE** (Roe et al., 2005), **MIMIC-IV** (Johnson et al., 2023). For these datasets, during pre-training, we construct semi-synthetic classification tasks where labels are sampled from Bayesian Neural Network (BNN) priors (Müller et al., 2021). The feature distributions for the training tasks are obtained by sampling real instances and introducing synthetic missingness. We then evaluate the model performance on test datasets, constructed with varying sizes and degrees of missingness, where the labels are obtained from the original datasets. This setup preserves realistic feature distributions, enables controlled evaluation across varying degrees of missingness and sample sizes, and assesses model generalization to tasks with real labels.

To further demonstrate our model's capability on real datasets, we evaluate **L2M** on **MNIST**, where training task labels are drawn from real binary digit-pair tasks rather than semi-synthetic priors. Images are divided into $d = 20$ candidate pixel blocks for acquisition. At each training step, we sample

a binary classification task between two randomly chosen digits, training the model to adaptively differentiate between images of two digits in-context. We evaluate the performance on datasets with varying missingness and sample sizes, using unseen queries.

**Baselines** Because prior AFA work is task-specific, we compare against AFA methods that train a separate model for each task. We focus on two greedy, CMI-based approaches: gradient dynamic feature selection (**GDFS** (Covert et al., 2023)), which uses MLPs instead of sequence modeling, and discriminative mutual information estimation (**DIME** (Gadgil et al., 2023)), which trains an MLP as a value network to estimate CMI directly. To ensure fair evaluation, both **L2M** and task-specific models are evaluated on the same held-out tasks and query sets. We also include an RL baseline using Deep Q-learning (**DQN**) (Shim et al., 2018; Kachuee et al., 2019; Janisch et al., 2019), but exclude it from synthetic tasks due to the prohibitive computational cost of learning hundreds of test tasks. To ensure comparability with greedy strategies, we define the per-step reward as the log likelihood under the current predictor, and each trajectory terminates when all features are acquired.

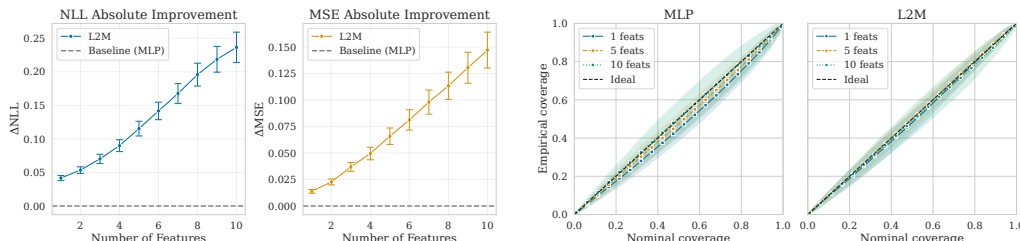

Figure 3: Left: Improvement in uncertainty estimation relative to baseline task-specific MLP (*higher is better*). The **L2M** approach shows progressively increasing gains as more features are acquired in the trajectory. Right: Coverage plots comparing MLP and **L2M** at various acquisition steps. **L2M** also shows robust coverage compared to the MLP at various acquisition steps. Error represents standard error across 200 sampled evaluation tasks. To ensure a fair comparison, both methods are evaluated on the same random acquisition trajectories using consistent evaluation tasks and samples.

**Results** Figure 3 shows the absolute improvement of **L2M** over task-specific MLP baselines in uncertainty quantification on synthetic GP tasks, evaluated by log loss and mean squared error (MSE). Corresponding semi-synthetic and real-world task results are in Appendix Figure 7. Overall, **L2M** provides more reliable uncertainty estimates, with larger gains in log loss and MSE as more features are acquired. This phenomenon likely arises because, at later acquisition steps, retrospective missingness reduces joint data coverage for the required feature sets, which makes MLPs struggle to learn. In contrast, **L2M** leverages its learned prior to mitigate reduced coverage. Improved uncertainty quantification is particularly important, as it directly translates to better downstream feature acquisition performance as shown in both the RBF and Matern kernel tasks in Figure 4.

Figure 4 demonstrates improved log loss of **L2M** over all relevant baselines on evaluation tasks across datasets (additional metrics are shown in Appendix Figure 9). The magnitude of **L2M**'s gains over baselines varies by dataset, depending on how well the pretraining task prior aligns with the downstream task. Our adaptive strategies offer only marginal gains over random acquisition on some real datasets. We attribute this to high task complexity and limited training data, which together limit the benefits of adaptivity. Nonetheless, the benefit of reliable uncertainty quantification, particularly leveraging pretraining on diverse tasks, is clear compared to task-specific AFA. The performance of task-specific AFA baselines deteriorate as more features are acquired, due to difficulties in learning a predictor and policy with limited data. Figure 5 demonstrates that **L2M** delivers the largest benefits in settings with shorter contexts and higher rates of retrospective missingness. Autoregressive meta-training via sequence loss enables reliable propagation of uncertainty at different context (dataset) sizes, leading to robustness at different context lengths. Furthermore, we hypothesize that robust performance across varying rates of retrospective missingness arises from our sequence modeling framework that both explicitly represents and effectively propagates the uncertainty arising from missingness in the historical data. This is especially useful in healthcare, where labeled data may be limited, and certain measurements may exhibit high rates of retrospective missingness.

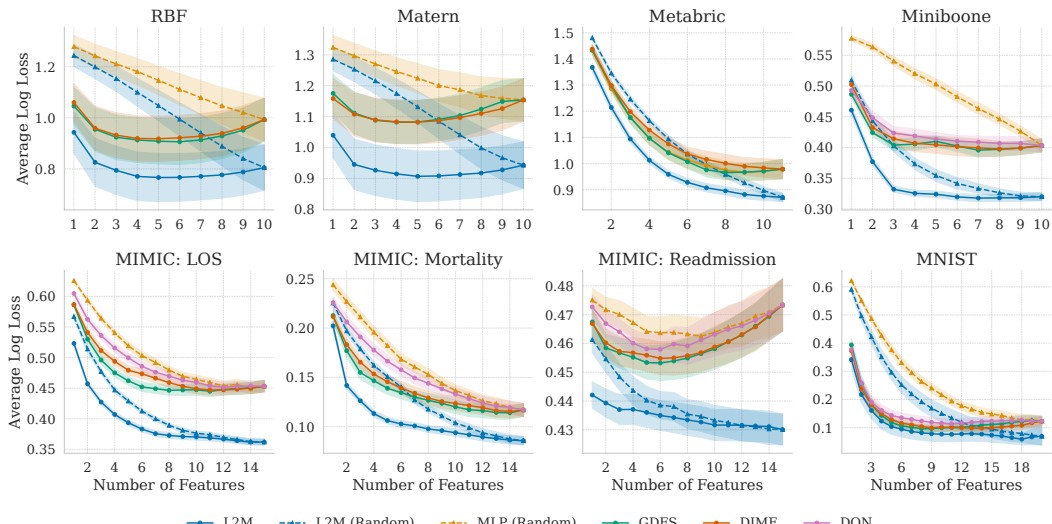

Figure 4: Acquisition performance quantified by log loss averaged over tasks derived from various synthetic and real-world datasets. MIMIC-IV demonstrate the ability for a single pretrained L2M model to generalize to diverse tasks with real unseen labels. The acquisition performance also often outperforms task-specific greedy and RL approaches.

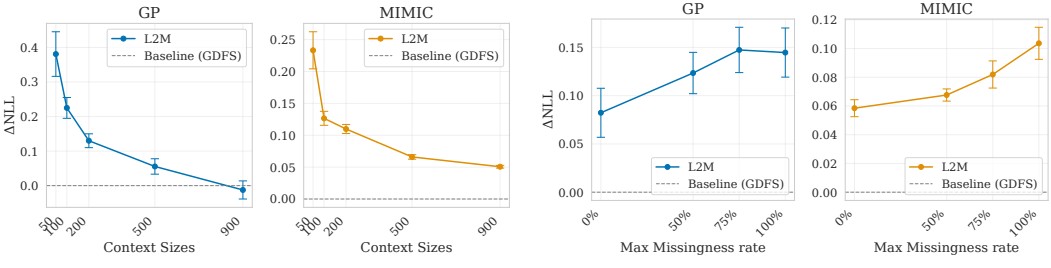

(a) NLL improvement across context lengths      (b) NLL improvement across missingness rates

Figure 5: Average improvement in log loss (y-axis) for the GP tasks and MIMIC-LOS for a fixed set of evaluation tasks while varying the number of context samples, and levels of retrospective missingness in each task (x-axis). Task-specific improvements are averaged over 100 (GP) and 50 (MIMIC) tasks. **L2M** is robust to settings with fewer shots (labeled samples) and with higher rates of feature missingness. **L2M** also achieves length generalization for the GP task, as it is only trained on sequences of 500 samples.

## 6   DISCUSSION

In this work, we formulate the meta-AFA problem and present an end-to-end differentiable uncertainty-driven approach for greedy feature acquisition that performs in-context learning across tasks. We show significant improvement using **L2M** in benchmarks and realistic healthcare datasets, demonstrating robustness under limited labeled data or significant retrospective missingness.

**Limitations** *(i)* Our approach relies on sufficient offline action coverage and MAR, an untestable but realistic assumption about the missingness mechanism. Future work will relax these assumptions and investigate whether our uncertainty estimates can serve as informative bounds or diagnostics for violations of positivity or MAR (Jesson et al., 2020). *(ii)* Empirically, we demonstrate the utility with tabular models pretrained from scratch on simple synthetic task priors. Scaling to diverse, large-scale real datasets across domains is deferred to future work, highlighting the need for principled prior-specification procedures to enable scalability and broad applicability. Leveraging priors encoded in pretrained language models is another promising direction. *(iii)* We focus on learning greedy one-

step acquisition policies and therefore do not consider multi-step planning for long-term reward. We discuss limitations of greedy acquisition in Appendix A.3 and leave planning with in-context reinforcement learning (Moeini et al., 2025) to future work. *(iv)* While we demonstrate proof-of-concept on medium-scale tabular datasets, extension to large tabular datasets is a crucial engineering challenge. *(v)* Finally, we restrict attention to time-invariant settings; extending to time-varying dynamics is a crucial aspect of future work.

**Ethics Statement** Our paper is a technical proof-of-concept. While we demonstrate evaluations in healthcare, additional evaluations regarding fairness and generalizability are necessary before this method is deployed in the real world.

**Reproducibility Statement** We evaluate on publicly available datasets. All code to reproduce our experiments will be publicly available, and a link to anonymous source code is provided in Appendix A.6.8. Details of the experimental design and hyperparameters are outlined in Appendix A.6.

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

# A  APPENDIX

## A.1  PROOF OF THEOREM 3.4

**Theorem 3.4** The CMI objective under missing data is given by

$$I(Y; X_j(1) \mid \underline{X}_t)$$

$$= \sum_{Y, X_j(1)} p(Y, X_{t+1}(1) \mid \underline{X}_t) \log \frac{p(Y, X_j(1) \mid \underline{X}_t)}{p(Y \mid \underline{X}_t)p(X_j(1) \mid \underline{X}_t)}$$

$$= \sum_{Y, X_j(1)} p(Y \mid X_j(1), \underline{X}_t)p(X_j(1)|\underline{X}_t) \log \frac{p(Y \mid X_j(1), \underline{X}_t)p(X_j(1)|\underline{X}_t)}{p(Y \mid \underline{X}_t)p(X_j(1) \mid \underline{X}_t)}$$

$$= \sum_{Y, X_j(1)} p(Y \mid X_j(1), \underline{X}_t, R_j = 1)p(X_j(1)|\underline{X}_t, R_{t+1} = 1) \log \frac{p(Y \mid X_j(1), \underline{X}_t, R_j = 1)p(X_j(1)|\underline{X}_t, R_j = 1)}{p(Y \mid \underline{X}_t)p(X_j(1) \mid \underline{X}_t)}$$

$$= \sum_{Y, X_j} p(Y \mid X_j, \underline{X}_t, R_j = 1)p(X_j|\underline{X}_t, R_j = 1) \log \frac{p(Y \mid X_j, \underline{X}_t, R_j = 1)p(X_j|\underline{X}_t, R_j = 1)}{p(Y \mid \underline{X}_t)p(X_j \mid \underline{X}_t)}$$

$$= \sum_{Y, X_j} p(Y, X_j \mid \underline{X}_t, R_j = 1) \log \frac{p(Y, X_j \mid \underline{X}_t, R_j = 1)}{p(Y \mid \underline{X}_t)p(X_j \mid \underline{X}_t, R_j = 1)}$$

$$= I(Y; X_j \mid \underline{X}_t, R_j = 1),$$

The third equality holds due to the MAR assumption $R_j \perp\!\!\!\perp X_j(1)|\underline{X}_t$ and exclusion restriction (no direct measurement effect) $R_j \perp\!\!\!\perp Y|X_j(1), \underline{X}_t$. Positivity ensures all conditional distributions are well-defined.

## A.2  PROOF OF PROPOSITION A.2

We begin by showing that minimizing the surrogate one-step loss given in Equation 7 for a single task recovers the greedy CMI actions. Our result is a slight modification from the result shown in (Covert et al., 2023).

The surrogate loss for a single task $\mathcal{T} \sim p_{\mathcal{T}}$ is given by:

$$\mathcal{L}(\theta, \phi) = \mathbb{E}_{S_t, A_t \sim \tilde{\pi}(\cdot|S_t), R}\left[\mathbb{E}_{X_{A_t}, Y \mid \underline{X}_t, R_{A_t}=1}\left[\ell(f_\phi(\cdot \mid \underline{X}_t \cup X_{A_t}), Y)\right]\right] \tag{7}$$

where states are denoted as $S_t = (\underline{X}_t, \underline{A}_t)$ and the acquisition action is sampled from the blocked policy $A_t \sim \tilde{\pi}_\theta(\cdot \mid S_t)$.

*Remark* A.1. For this theorem, the loss can average over the state distribution at step $t$, denoted $d_t^{\tilde{\pi}_\theta}(\cdot \mid \mathcal{T})$, obtained by rolling out the (blocked) policy $\tilde{\pi}_\theta$ for $t$ steps from the initial law $d_0(\cdot \mid \mathcal{T})$ induced by $p_{\mathcal{T}}(X, R, Y)$. The choice of $d_t$ is flexible (with caveats) and should have sufficient overlap with the intended deployment state distribution. Note that the per-state argmax (the optimal acquisition action at each step) does not change regardless of the outer state distribution.

**Proposition A.2.** *(Surrogate optimality for greedy CMI) For a given task $\mathcal{T}$, consider the population objective (Eq. 7 with cross-entropy loss). Then any joint minimizer $(\theta^\star, \phi^\star)$ of $\mathcal{L}$ satisfies:*

1. *$\phi^\star$ is Bayes-optimal: $f_{\phi^\star}(\cdot \mid \underline{X}_t) = p_{\mathcal{T}}(Y \mid \underline{X}_t)$;*

2. *$\pi_{\theta^\star}$ places all its mass on actions that maximize the conditional mutual information*

$$j \in \arg\max_{j:R_j=1} I_{\mathcal{T}}\big(Y; X_j \mid \underline{X}_t = \underline{x}_t, R_j = 1\big),$$

   *i.e. $\pi_{\theta^\star}(j \mid \underline{x}_t) > 0$ only if $j$ is a greedy-CMI maximizer. If the maximizer is unique, $\pi_{\theta^\star}$ is a point mass on that action.*

*Proof.* Part I - Proof of Bayes-optimality:

We fix $\theta$ and consider the predictor. We begin with a standard fact: under cross-entropy loss for a discrete binary outcome, the conditional risk is minimized by the true conditional. In other words, to minimize expected loss the model $f_\phi$ needs to closely approximate the true distribution $p$. We show that this holds agnostic to the choice of $\mathcal{T}$.

**Lemma A.3** (Bayes optimality under cross-entropy). *Let $\ell(q, y) = -\log q(y)$. Then the minimizer $\phi^\star$ satisfies*

$$f_{\phi^\star}(\underline{X}_t) = \arg\min_{f_\phi(\cdot)} \mathbb{E}_{Y|\underline{X}_t}[\ell(f_\phi(\cdot \mid \underline{X}_t), Y)] = p(Y \mid \underline{X}_t).$$

*Furthermore, this minimizer does not depend on $\theta$. In particular, any $f_{\phi^\star}$ that matches the true conditional for all such $(\underline{x}_t, x_j)$ is a global minimizer for every policy.*

*Proof.* We denote $p(i \mid \underline{X}_t) = p_i$ as the conditional class probabilities and $f_i = f_\phi(i \mid \underline{X}_t)$ as the learner's predicted class probabilities, where $i \in \{0, 1\}$ are the binary class labels. The conditional risk decomposes as

$$\mathbb{E}_{Y|\underline{X}_t}[\ell(f_\phi(\cdot \mid \underline{X}_t), Y)] = -\sum_{i=0}^{1} p_i \log f_i$$

$$= -\sum_{i=0}^{1} p_i \log p_i + \sum_{i=0}^{1} p_i \log \frac{p_i}{f_i} = \underbrace{H(Y \mid \underline{X}_t)}_{\text{Constant}} + \text{KL}(p(Y|\underline{X}_t) \,\|\, f_\phi(Y|\underline{X}_t)).$$

$\square$

Part II - Proof of maximizer equivalence

Once we have Bayes optimality with the learner at $\phi^\star$, we can rewrite the inner risk as the expected conditional entropy using the following lemma:

**Lemma A.4** (Risk reduces to (expected) conditional entropy at $\phi^\star$). *With $\ell(q, y) = -\log q(y)$ and $f_{\phi^\star}$ as in Lemma A.3, for any task $\mathcal{T}$ and step $t$, any history $\underline{x}_t$ and retrospective feature availability $r_j$*

$$\mathbb{E}_{Y, X_j | \underline{x}_t, R_j = r_j}[\ell(f_{\phi^\star}(\cdot \mid \underline{x}_t \cup X_j), Y)] = \mathbb{E}_{X_j | \underline{x}_t, R_j = r_j}\big[H_{\mathcal{T}}(Y \mid \underline{x}_t, X_j)\big].$$

*Consequently, the policy-evaluated inner term in equation 7 is*

$$\mathbb{E}_{A_t \sim \tilde{\pi}_\theta(\cdot|s_t)} \mathbb{E}_{X_{a_t} | \underline{x}_t, R_{a_t} = 1}\big[H_{\mathcal{T}}(Y \mid \underline{x}_t, X_{a_t})\big].$$

*Proof.* Performing the similar decomposition as in Lemma A.3,

$$\mathbb{E}_{Y, X_j | \underline{x}_t, r_j}\big[\ell(f_{\phi^\star}(\cdot \mid \underline{x}_t \cup X_j), Y)\big]$$

$$= \mathbb{E}_{X_j | \underline{x}_t, r_j}\Big[\mathbb{E}_{Y | \underline{x}_t, X_j}\big[-\log f_{\phi^\star}(Y \mid \underline{x}_t, X_j)\big]\Big]$$

$$= \mathbb{E}_{X_j | \underline{x}_t, r_j}\Big[H_{\mathcal{T}}(Y \mid \underline{x}_t, X_j) + \text{KL}\big(p_{\mathcal{T}}(Y \mid \underline{x}_t, X_j) \,\|\, f_{\phi^\star}(Y \mid \underline{x}_t, X_j)\big)\Big]$$

$$= \mathbb{E}_{X_j | \underline{x}_t, r_j}\big[H_{\mathcal{T}}(Y \mid \underline{x}_t, X_j)\big]$$

Where the first equality follows from iterated expectations. In the last equality, the KL term vanishes at $\phi^\star$. $\square$

Now, we consider the loss in equation 7. Plugging $\phi^\star$ and using Lemma A.4 for the given choice of task $\mathcal{T}$,

$$\mathcal{L}(\theta, \phi^\star) = \mathbb{E}_{S_t, R}\Bigg[\mathbb{E}_{A_t \sim \tilde{\pi}_\theta(\cdot|s_t)} \mathbb{E}_{X_{a_t} | \underline{x}_t, R_{a_t} = 1}\big[H_{\mathcal{T}}(Y \mid \underline{x}_t, X_{a_t})\big]\Bigg]$$

$$\overset{\text{blocked}}{=} \sum_{s_t} \sum_{r \in \{0,1\}^d} p(s_t, r)\Bigg[\sum_{\{a_t : R_{a_t} = 1\}} \tilde{\pi}_\theta(a_t \mid s_t) \mathbb{E}_{X_{a_t} | \underline{x}_t, R_{a_t} = 1}\big[H_{\mathcal{T}}(Y \mid \underline{x}_t, X_{a_t})\big]\Bigg]$$

where $\{a_t : R_{a_t} = 1\}$ is the set of available features at step $t$. For fixed $\underline{x}_t$,

$$\sum_{a_t \in \{a_t : R_{a_t} = 1\}} \pi(a_t | s_t) \, \mathbb{E}_{X_{a_t} | \underline{x}_t, \, R_{a_t} = 1}[H_{\mathcal{T}}(Y \mid \underline{x}_t, X_{a_t})]$$

is linear over the simplex on $\{a_t : R_{a_t} = 1\}$ and is therefore minimized by placing all mass on

$$\arg\min_{a_t \in \{a_t : R_{a_t} = 1\}} \mathbb{E}_{X_{a_t} | \underline{x}_t, \, R_{a_t} = 1}[H_{\mathcal{T}}(Y \mid \underline{x}_t, X_{a_t})] .$$

Since $H(Y \mid \underline{x}_t)$ does not depend on $a_t$,

$$\arg\min_{a_t \in \{a_t : R_{a_t} = 1\})} \mathbb{E}_{X_{a_t} | \underline{x}_t, \, R_{a_t} = 1}[H_{\mathcal{T}}(Y \mid \underline{x}_t, X_{a_t})] = \arg\max_{a_t \in \{a_t : R_{a_t} = 1\})} I_{\mathcal{T}}(Y; X_{a_t} \mid \underline{x}_t, R_{a_t} = 1) ,$$

because

$$I_{\mathcal{T}}(Y; X_{a_t} \mid \underline{x}_t, R_{a_t} = 1) = H_{\mathcal{T}}(Y \mid \underline{x}_t) - \mathbb{E}_{X_{a_t} | \underline{x}_t, \, R_{a_t} = 1}[H_{\mathcal{T}}(Y \mid \underline{x}_t, X_{a_t})] .$$

If the maximizer is unique, the minimizer $\pi_{\theta^\star}(\cdot \mid s_t)$ is a point mass on that action. $\qquad \square$

### A.3 Proof of Theorem 4.2

Our main theorem is an extension of Proposition A.2 to a Bayesian setting, which we approximate using sequence models.

We leverage the following conditional independence assumption, which improves tractability by removing the need to fully model the joint via an autoregressive factorization.

**Assumption A.5** (Conditional independence across queries)**.** Given the context $Z^{1:m}$ and the per-query partial inputs $\underline{X}_t^{m+1:N}$, the query points are conditionally independent:

$$p\big(Y^{m+1:N} \mid \underline{X}_t^{m+1:N}, Z^{1:m}\big) = \prod_{q=m+1}^{N} p\big(Y^{(q)} \mid \underline{X}_t^{(q)}, Z^{1:m}\big) .$$

**Theorem 4.2**[Surrogate optimality for greedy CMI with context] Consider the sequence modeling objective in Eq. 5 with cross-entropy loss. Let $Z^{1:m} = (X^{1:m}, R^{1:m}, Y^{1:m})$ be the task-specific context and $\underline{X}_t^{m+1:N}$ the partially observed features for the $N - m$ query points at step $t$. Then any joint minimizer $(\theta^\star, \phi^\star)$ of $\mathcal{L}$ satisfies:

1. **Per-query Bayes-optimality.** For each $q \in \{m+1, \ldots, N\}$,

$$f_{\phi^\star}\big(\cdot \mid \underline{X}_t^{(q)}, Z^{1:m}\big) = p\big(Y^{(q)} \mid \underline{X}_t^{(q)}, Z^{1:m}\big) .$$

   Consequently by assumption A.5 ,

$$f_{\phi^\star}\big(\cdot \mid \underline{X}_t^{m+1:N}, Z^{1:m}\big) = p\big(Y^{m+1:N} \mid \underline{X}_t^{m+1:N}, Z^{1:m}\big)$$

2. **Step-wise CMI-optimal acquisition for every query.** For each query index $q \in \{m+1, \ldots, N\}$ and for $(\underline{x}_t^{(q)}, z^{1:m})$, the policy places mass only on actions that maximize CMI:

$$j \in \arg\max_{a_t : R_{a_t} = 1} I\Big(Y^{(q)}; X_{a_t}^{(q)} \,\Big|\, \underline{X}_t^{(q)} = \underline{x}_t^{(q)}, R_{a_t}^{(q)} = 1, Z^{1:m} = z^{1:m}\Big) .$$

   If the maximizer is unique, $\pi_{\theta^\star}(\cdot \mid \underline{x}_t^{(q)}, z^{1:m})$ is a point mass on that action.

Part I - Proof of Bayes-optimality:

We first fix $\theta$ and consider the predictor. The loss is given by

$$\mathcal{L}(f_\phi, \pi_\theta) = \sum_{\mathcal{T} \sim p(\mathcal{T})} \sum_{m=1}^{N-1} \mathbb{E}_{S^{m+1}, \, Z^{1:m}, \, R}\left[ \mathbb{E}_{A_t \sim \tilde{\pi}(\cdot | S_t^{m+1}, Z^{1:m})}[J(A_t; \underline{X}_t^{m+1}, Z^{1:m})] \right].$$

where

$$J(a_t; \underline{x}_t^{m+1}, z^{1:m}) = \mathbb{E}_{\substack{X_{a_t}^{m+1}, Y^{m+1} | \underline{x}_t^{m+1}, \\ z^{1:m}, R_{a_t}=1}} \left[ \ell\big(f_\phi(\cdot \mid \underline{x}_t^{m+1} \cup X_{a_t}^{m+1}, z^{1:m}), Y^{m+1}\big) \right]$$

is the per-action expected loss.

We show [1] by showing that to minimize expected loss, $f_\phi$ needs to closely approximate the true distribution $p$, analogous to Lemma A.3.

*Proof.* Without loss of generality, we fix the context length $m$. We consider the minimizer $\phi^\star$ of the loss summed over each query $q \in \{m+1, ..., N\}$. Lemma A.3 applied to each query shows that this loss recovers the per-query conditional i.e.

$$f_{\phi^\star}\big(\cdot \mid \underline{X}_t^{(q)}, Z^{1:m}\big) = p\big(Y^{(q)} \mid \underline{X}_t^{(q)}, Z^{1:m}\big).$$

We now show that the loss minimizer also recovers the joint conditional

$$\sum_{q=m+1}^{N} \mathbb{E}_{Y^{(q)} | \underline{X}_t^{(q)}, Z^{1:m}} \left[ \ell\big(f_{\phi^\star}(\cdot \mid \underline{X}_t^{(q)}, Z^{1:m}), Y^{(q)}\big) \right]$$

$$\overset{\text{change of measure}}{=} \sum_{q=m+1}^{N} \mathbb{E}_{Y^{(q)} | \underline{X}_t^{m+1:N}, Z^{1:m}} \left[ - \frac{p\big(Y^{(q)} \mid \underline{X}_t^{(q)}, Z^{1:m}\big)}{p\big(Y^{(q)} \mid \underline{X}_t^{m+1:N}, Z^{1:m}\big)} \log f_{\phi^\star}(Y^{(q)} \mid \underline{X}_t^{(q)}, Z^{1:m}) \right]$$

$$\overset{\text{linearity of } \mathbb{E}}{=} \mathbb{E}_{Y^{m+1:N} | \underline{X}_t^{m+1:N}, Z^{1:m}} \left[ - \sum_{q=m+1}^{N} \frac{p\big(Y^{(q)} \mid \underline{X}_t^{(q)}, Z^{1:m}\big)}{p\big(Y^{(q)} \mid \underline{X}_t^{m+1:N}, Z^{1:m}\big)} \log f_{\phi^\star}(Y^{(q)} \mid \underline{X}_t^{(q)}, Z^{1:m}) \right]$$

$$\overset{\text{CI assumption A.5}}{=} \mathbb{E}_{Y^{m+1:N} | \underline{X}_t^{m+1:N}, Z^{1:m}} \left[ - \sum_{q=m+1}^{N} \log f_{\phi^\star}\big(Y^{(q)} \mid \underline{X}_t^{(q)}, Z^{1:m}\big) \right]$$

$$\overset{\text{log manipulation}}{=} \mathbb{E}_{Y^{m+1:N} | \underline{X}_t^{m+1:N}, Z^{1:m}} \left[ - \log \prod_{q=m+1}^{N} f_{\phi^\star}\big(Y^{(q)} \mid \underline{X}_t^{(q)}, Z^{1:m}\big) \right]$$

$$\overset{\text{factorized predictor}}{=} \mathbb{E} \left[ - \log f_{\phi^\star}\big(Y^{m+1:N} \mid \underline{X}_t^{m+1:N}, Z^{1:m}\big) \right]$$

$$= H\big(Y^{m+1:N} \mid \underline{X}_t^{m+1:N}, Z^{1:m}\big) + \text{KL}\Big(p\big(Y^{m+1:N} \mid \underline{X}_t^{m+1:N}, Z^{1:m}\big) \,\big\|\, f_\phi\big(Y^{m+1:N} \mid \underline{X}_t^{m+1:N}, Z^{1:m}\big)\Big)$$

$$\square$$

Part II - Proof of CMI-optimal acquisition

*Proof.* We consider the loss in equation 5, and plug $\phi^\star$ in and use Lemma A.4 for the sequence scenario.

We first rewrite the per-action expected loss in terms of the conditional entropy.

$$J(a_t; \underline{x}_t^{m+1}, z^{1:m}, \phi^\star) = \mathbb{E}_{\substack{X_{a_t}^{m+1}, Y^{m+1} | \underline{x}_t^{m+1}, \\ z^{1:m}, R_{a_t}=1}} \left[ \ell\big(f_{\phi^\star}(\cdot \mid \underline{x}_t^{m+1} \cup X_{a_t}^{m+1}, z^{1:m}), Y^{m+1}\big) \right]$$

$$= \mathbb{E}_{\substack{X_{a_t}^{m+1}, | \underline{x}_t^{m+1}, \\ z^{1:m}, R_{a_t}=1}} \left[ H(Y^{m+1} \mid X_{a_t}^{m+1}, x_t^{m+1}, z^{1:m}) \right]$$

Now we use the same logic as in Theorem A.2. Plugging the per-action expected loss back into the total loss:

$$\mathcal{L}(f_\phi, \pi_\theta) = \sum_{\mathcal{T} \sim p(\mathcal{T})} \sum_{m=1}^{N-1} \mathbb{E}_{S_t^{m+1}, Z^{1:m}, R} \left[ \mathbb{E}_{A_t \sim \tilde{\pi}_\theta(\cdot | S_t^{m+1}, Z^{1:m})} [J(A_t; \underline{X}_t^{m+1}, Z^{1:m}, \phi^\star)] \right]$$

$$= \sum_{\mathcal{T} \sim p(\mathcal{T})} \sum_{m=1}^{N-1} \sum_{s_t^{m+1}, z^{1:m}} \sum_{r \in \{0,1\}^d} p(s_t^{m+1}, z^{1:m}, r) \left[ \sum_{a_t^m : r_{a_t} = 1} \tilde{\pi}_\theta(a_t^m \mid s_t^{m+1}, z^{1:m}) J(a_t^m; \underline{x}_t^{m+1}, z^{1:m}, \phi^\star) \right].$$

Therefore, for each tuple $(s_t^{m+1}, z^{1:m}, r)$, the inner summation

$$\sum_{a_t^m : r_{a_t} = 1} \tilde{\pi}_\theta(a_t^m \mid s_t^{m+1}, z^{1:m}) J(a_t^m; \underline{x}_t^{m+1}, z^{1:m}, \phi^\star)$$

is linear over the simplex on $\{a_t^m : r_{a_t} = 1\}$ and is therefore minimized when we select the acquisition action $\arg\min_{a_t^m} J(a_t^m; \underline{x}_t^{m+1}, z^{1:m}, \phi^\star)$. Therefore, the loss minimizer $\tilde{\pi}_{\theta^\star}$ places all mass on the CMI optimal action.

$\square$

### A.4 DISCUSSION ON GREEDY ACQUISITION

We provide additional discussion on greedy acquisition using the concept of adaptive submodularity (Golovin & Krause, 2011). Let there be $d$ available features with index set $[d] := \{1, \ldots, d\}$. A full realization is $x \in \mathcal{X}^d$, where the value of feature $j$ is $x_j \in \mathcal{X}$. We aim to maximize a nonnegative utility $f : 2^{[d]} \times \mathcal{X}^d \to \mathbb{R}_{\geq 0}$, where $g(S, x)$ evaluates the utility of acquiring the subset $S \subseteq [d]$ under realization $x$.

We begin by recalling submodularity.

**Definition A.6.** A set function $g : 2^{[d]} \to \mathbb{R}$ is *submodular* if for all $A \subseteq B \subseteq [d]$ and every $j \in [d] \setminus B$,

$$g(A \cup \{j\}) - g(A) \geq g(B \cup \{j\}) - g(B).$$

Intuitively, the property of submodularity implies diminishing returns. We now recall the definition of adaptive submodularity for feature acquisition

**Definition A.7.** Let $X = (X_1, \ldots, X_d) \in \mathcal{X}^d$ be random. A utility $f : 2^{[d]} \times \mathcal{X}^d \to \mathbb{R}_{\geq 0}$ is *adaptively submodular* if for all sets $S \subseteq S' \subseteq [d]$, for all indices $j \in [d] \setminus S'$, and for all partial realizations $x_S$ and $x_{S'}$, the conditional expected marginal benefit does not increase as more outcomes are observed:

$$\Delta(j \mid x_S) = \mathbb{E}_{X|x_s} [f(S \cup \{j\}, X) - f(S, X)]$$
$$\geq \mathbb{E}_{X|x_{s'}} [f(S' \cup \{j\}, X) - f(S', X)] = \Delta(j \mid x_{S'}).$$

The theoretical result in (Golovin & Krause, 2011) shows that for a fixed budget $k$, the greedy policy for a distribution that satisfies definition A.7 (and adaptive monotone) achieves an $(1 - e^{-1})$ approximation to the expected reward of the best policy, following from the result that the optimality gap shrinks by an $(1 - k^{-1})$ factor at each step. However, greedy CMI-based acquisition is not adaptively submodular for problems where features are jointly informative. For example, feature $j$ (a chest X-ray) may only be informative after a different feature has been observed due to synergistic information (an electrocardiogram), but uninformative on its own. (Norcliffe et al., 2025) provides an intuitive example using an indicator variable that determines which features are informative, and also discusses limitations of CMI if the objective is 0-1 loss minimization.

### A.5 MODEL ARCHITECTURE AND TRAINING DETAILS

The goal is to model the one-step predictive distributions $p(Y^{(q)} \mid \underline{X}_t^{(q)} = \underline{x}_t^{(q)}, Z^{1:m} = z^{1:m})$, also referred to as posterior predictive distributions (PPD). We refer to various previous works for

formalizing the connection between sequence modeling and Bayesian inference (Müller et al., 2021; Nguyen & Grover, 2022; Ye & Namkoong, 2024). We leverage the insight that the sequence model for performing explicit Bayesian inference must satisfy the following inductive invariances (Nguyen & Grover, 2022; Ye & Namkoong, 2024).

### A.5.1 MODEL ARCHITECTURE

**Definition A.8. Context Invariance.** A model $f_\phi$ is context invariant if for any choice of permutation function $\pi$ and $m \in [1, N-1]$, $f_\phi(Y^{m+1:N} | \underline{X}_t^{m+1:N}, Z^{1:m}) = f_\phi(Y^{m+1:N} | \underline{X}_t^{m+1:N}, Z^{\pi(1):\pi(m)})$

**Definition A.9. Target Equivariance.** A model $f_\phi$ is target equivariant if for any choice of permutation function $\pi$ and $m \in [1, N-1]$, $f_\phi(Y^{m+1:N} | \underline{X}_t^{m+1:N}, Z^{1:m}) = f_\phi(Y^{\pi(m+1):\pi(N)} | \underline{X}_t^{\pi(m+1):\pi(N)}, Z^{1:m})$

We approximate these invariances using a Transformer model (Vaswani et al., 2017) with several modifications. For each input query sample $i$, the sufficient statistics for the state $s_t^i$ are the partially observed feature values $\underline{x}_t^i \in \mathbb{R}^d$ together with the acquisition mask $\underline{a}_{t-1}^i \in \{0,1\}^d$, which records which features have been acquired so far. The state is encoded by applying the mask to the feature vector, $x_t^i \odot \underline{a}_{t-1}^i$, and concatenating this with the mask itself. Finally, we append a zero vector of length $c$ to represent the unobserved target outcome $Y$. The resulting input representation for a query sample is

$$z_{\text{qry}}^i = \begin{bmatrix} x_t^i \odot \underline{a}_{t-1}^i, & \underline{a}_{t-1}^i, & \mathbf{0}^c \end{bmatrix} \in \mathbb{R}^{2d+c}.$$

For each context sample $i$, the sufficient statistics consist of the partially observed feature values $x^i \in \mathbb{R}^d$ together with the retrospective missingness mask $r^i \in \{0,1\}^d$, which indicates which features were collected in the past. We append the observed target outcome $y^i$ to form the encoded representation.

$$z_{\text{ctx}}^i = \begin{bmatrix} x^i \odot r^i, & r^i, & y^i \end{bmatrix} \in \mathbb{R}^{2d+c}.$$

Next, we remove standard positional embeddings and replace the usual causal attention mask with a custom design, since causal masking does not satisfy the invariances in Definition A.8. To enable efficient computation of the autoregressive loss, we also introduce *target points* into the sequence during training.

Each input sequence for autoregressive loss computation has length $2N - m$ and is ordered as

$$\{ z_{\text{ctx}}^1, \ldots, z_{\text{ctx}}^m, z_{\text{tar}}^{m+1}, \ldots, z_{\text{tar}}^N, z_{\text{qry}}^{m+1}, \ldots, z_{\text{qry}}^N \}.$$

Each target point $z_{\text{tar}}^i$ shares the same underlying feature vector $x^i$ as its corresponding query $z_{\text{qry}}^i$, but encodes the retrospective mask $r^i$ and observed outcome $y^i$ in place of zero-padding.

The attention mask 6 enforces the following structure:

- Context points can attend freely to one another.
- Each target point can attend to all context points and all preceding target points.
- Each query point $z_{\text{qry}}^i$ can attend to context points and preceding target points, but not to other queries.

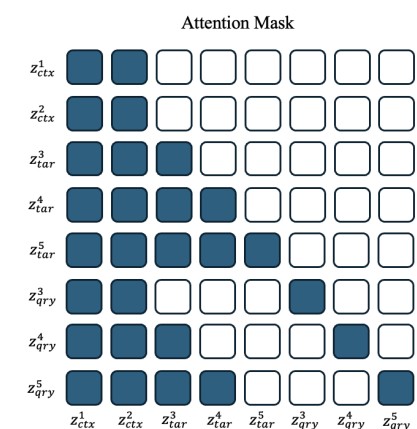

Figure 6: Attention mask used during training with 2 context samples and 3 query samples. Each query has a paired target sample that shares the same features but includes the observed outcome $y$ and retrospective mask $r$ instead of zero-padding.

### A.5.2 TRAINING

We provide the algorithm for pretraining the predictor in Algorithm 1 and pretraining the policy in Algorithm 2. The inference procedure is provided in Algorithm 3.

---

**Algorithm 1** Autoregressive training for sequence model $f_\phi$ given $p(\mathcal{T})$

---

**Input:** Predictor $f_\phi$ **Require:** Sequence length $N$, batch size $J$

1: **for until convergence do**
2:    **for** each task $\mathcal{D}_\mathcal{T} = \{X^{1:N}, Y^{1:N}, R^{1:N}\}$ in mini-batch **do**
3:       Initialize the set of observed indices $\underline{A}_0^{1:N}, \underline{A}_0 \subseteq [d]$ with always available feature indices
4:       **for** $t \in \{1,...,d-1\}$ **do**
5:         **for** $m \in \{1, ..., N-1\}$ **do**
6:           Predict next label using the sequence model:

$$\hat{Y}^{m+1} \sim f_\phi(\cdot \mid \underline{X}_t^{m+1}, X^{1:m}, R^{1:m}, Y^{1:m})$$

7:           Sample a random feature index $j : R_j^{m+1} = 1$ to acquire, so $\underline{A}_t^{m+1} \leftarrow \underline{A}_{t-1}^{m+1} \cup \{j\}$
8:         **end for**
9:       **end for**
10:    **end for**
11:    Compute mini-batch loss $\hat{l}_\phi$ and update parameters $\phi \leftarrow \phi - \eta \nabla_\phi \hat{l}_\phi$
12: **end for**
13: **return** trained model $\hat{f}_\phi$

---

---

**Algorithm 2** Autoregressive training for sequence model $\pi_\theta$ given $p(\mathcal{T})$

---

**Input:** Policy $\pi_\theta$, predictor $f_\phi$ **Require:** Sequence length $N$, batch size $J$

1: **for until convergence do**
2:   **for** each task $\mathcal{D}_\mathcal{T} = \{X^{1:N}, Y^{1:N}, R^{1:N}\}$ in mini-batch **do**
3:     Initialize the set of observed indices $\underline{A}_0^{1:N}, \underline{A}_0 \subseteq [d]$ with always available feature indices
4:     **for** $t \in \{1, ..., d-1\}$ **do**
5:       **for** $m \in \{1, ..., N-1\}$ **do**
6:         Given the state $S_t^{m+1} = (\underline{X}_t^{m+1}, \underline{A}_{t-1}^{m+1})$, output action distribution using policy:

$$\hat{A}^{m+1} \sim \pi_\theta(\cdot \mid S_t^{m+1}, X^{1:m}, R^{1:m}, Y^{1:m})$$

7:         Approximate argmax using straight through gumbel-softmax: $\tilde{A}^{m+1}$
8:         Compute and accumulate one-step loss using the predictor:

$$\ell\big(f_\phi(\cdot \mid \underline{X}_t^{m+1} \cup X_{\tilde{A}}, X^{1:m}, R^{1:m}, Y^{1:m}), Y^{m+1}\big)$$

9:         Sample a random feature index $j : R_j^{m+1} = 1$ to acquire, so $\underline{A}_t^{m+1} \leftarrow \underline{A}_{t-1}^{m+1} \cup \{j\}$
10:       **end for**
11:     **end for**
12:   **end for**
13:   Compute mini-batch loss $\hat{l}_\theta$ and update parameters $\theta \leftarrow \theta - \eta \nabla_\theta \hat{l}_\theta$
14: **end for**
15: **return** trained model $\hat{\pi}_\theta$

---

---

**Algorithm 3** Test-time inference procedure for solving an AFA task $\mathcal{T}$

---

**Require:** Pretrained sequence models $f_\phi, \pi_\theta$ **Input:** Samples from $\mathcal{D}_\mathcal{T} = \{X^{1:m}, R^{1:m}, Y^{1:m}\}$, and query samples $\underline{X}_0^{m+1:N}$ feature budget $k \leq d$

1: **for** $t \in \{1, \ldots, k\}$ **do**
2:   **for** $q \in \{m+1, \ldots, N\}$ **do**
3:     Compute $\pi_\theta(A_t^{(q)} \mid \underline{X}_t^{(q)}, \underline{A}_{t-1}^{(q)}, \mathcal{D}_\mathcal{T})$ and select action

$$a_t^i = \arg\max_a \ \pi_\theta(a \mid \underline{X}_t^{(q)}, \underline{A}_{t-1}^{(q)}, \mathcal{D}_\mathcal{T})$$

4:     Update $\underline{X}_{t+1}^{(q)} \leftarrow \underline{X}_t^{(q)} \cup X_{a_t}$ with chosen action
5:   **end for**
6: **end for**
7: **Return:** Predictions for all test samples in task $\mathcal{T}$

$$\hat{Y}^i \sim f_\phi\big(\cdot \mid \underline{X}_k^i, \mathcal{D}_\mathcal{T}\big), \quad \forall i \in \{m+1, \ldots, N\}$$

---

## A.6 EXPERIMENT DETAILS

### A.6.1 DATASETS

For each dataset, we construct two disjoint splits: a training set of size $n_{\text{train}}$ and a test pool of size $n_{\text{test}}$. For each dataset, we specify three components:

- **Baseline features** ($X_0$)**:** features that are always observed at the start of an acquisition trajectory.

- **Acquirable features** ($X_m$)**:** candidate features available for sequential acquisition.

- **Label space** ($Y$)**:** the outcome variable or class labels used for evaluation.

Table 1: Label prevalences (%) for final evaluation datasets.

| Dataset | % Positive |
|---|---|
| MiniBooNE | 72.16 |
| MIMIC-LOS | 44.90 |
| MIMIC-Readmission | 16.52 |
| MIMIC-Mortality | 8.67 |

1. **Metabric** ($n_{\text{train}} = 1{,}000$, $n_{\text{test}} = 898$):

$$X_m = \{\texttt{ccnb1, cdk1, e2f2, e2f7, stat5b, notch1,}$$
$$\texttt{rbpj, bcl2, egfr, erbb2, erbb3}\}.$$

$$Y \in \{\texttt{Luminal A, Luminal B, HER2-enriched,}$$
$$\texttt{Basal-like, Normal-like, Claudin-low}\}.$$

2. **MiniBooNE** ($n_{\text{train}} = 5{,}000$, $n_{\text{test}} = 10{,}000$):

$$X_m = \{\texttt{Feature 1, Feature 17, Feature 23, Feature 32, Feature 3,}$$
$$\texttt{Feature 27, Feature 12, Feature 4, Feature 25, Feature 2}\}.$$

$$Y \in \{\texttt{0}, \texttt{1}\}.$$

3. **MNIST** ($n_{\text{train}} = 30{,}000$, $n_{\text{test}} = 30{,}000$):

$$X_m = \{\texttt{block\_0\_4, block\_1\_6, block\_2\_2, block\_3\_5, block\_0\_3,}$$
$$\texttt{block\_0\_2, block\_5\_2, block\_4\_6, block\_5\_0, block\_4\_2,}$$
$$\texttt{block\_4\_3, block\_5\_6, block\_6\_3, block\_3\_3, block\_1\_3,}$$
$$\texttt{block\_5\_1, block\_4\_4, block\_3\_2, block\_5\_5, block\_2\_4}\}.$$

$$Y \in \{0, 1, 2, 3, 4, 5, 6, 7, 8, 9\}.$$

4. **MIMIC-IV** (Johnson et al., 2023) ($n_{\text{train}} = 5{,}000$, $n_{\text{test}} = 10{,}000$):

$$X_0 = \{\texttt{Age, Gender, ICU}\}, \quad X_m = \{\texttt{Hemoglobin, Platelet, RBC, WBC,}$$
$$\texttt{Bicarbonate, BUN, Calcium, Chloride, Creatinine, Glucose, RDW, INR,}$$
$$\texttt{PT, lymphocytes, monocytes}\}.$$

$$Y_{\text{Length of stay}} \in \{0, 1\},$$
$$Y_{\text{Mortality}} \in \{0, 1\},$$
$$Y_{\text{Readmission}} \in \{0, 1\}.$$

**Preprocessing** We define three binary classification tasks on **MIMIC-IV**. For each unique patient, we retain a single admission and set the prediction time to 48 hours after admission. Patients with an admission shorter than 48 hours are excluded. For each feature, we use the most recent measurement recorded before the prediction time; if no measurement is available from admission up to prediction time, the feature is treated as missing. To ensure that all ground-truth measurements are available for evaluation, we also exclude patients with missing values in any of the selected features in $X_m$. The tasks are defined as follows:

- **Length of stay (LOS):** whether the hospital stay extends at least 7 days beyond the prediction time.
- **Mortality:** whether the patient dies during the same hospital admission.
- **Readmission:** whether the patient is readmitted to the hospital within 30 days of discharge.

The tasks are generated using MEDS to facilitate reproducibility (Arnrich et al., 2024).

Table 2: Experimental setup across datasets. Each training step samples sequences of length $N$ from the pretraining pool. Feature values are normalized within each sequence.

| Dataset | Sequence length $N$ | Missingness | Task |
|---------|---------------------|-------------|------|
| **GP** | 500 | MCAR | Regression |
| **MiniBooNE** | 1000 | MCAR | Binary Classification |
| **MNIST** | 1000 | MCAR | Binary Classification |
| **Metabric** | 500 | MCAR | Multi-class Classification |
| **MIMIC-IV** | 1000 | MAR | Binary Classification |

### A.6.2 PRETRAINING TASK PRIOR

Here we describe the synthetic task prior used for pretraining our **L2M** models.

**GP.** We define a Gaussian process task prior with an RBF kernel to generate synthetic regression tasks. For each sampled task, we randomly select a subset of the input dimensions to be informative, while the remaining dimensions are treated as noise features. The kernel is parameterized with batch-specific lengthscales and output scales: lengthscales are drawn uniformly from the interval $[0.1, 5.0]$ for each dimension, and output scales are drawn uniformly from $[0.5, 2.0]$. Non-informative features are assigned a large lengthscale, effectively removing their contribution. An observation noise term $\sigma_\epsilon^2 I$ with $\sigma_\epsilon = 2 \times 10^{-2}$ is added for numerical stability.

**BNN.** We define a Bayesian neural network (BNN) task prior for classification tasks that generates synthetic labeling functions over feature inputs. For each sampled task, we proceed as follows:

1. **Selection of informative features.** For each batch, we randomly select a subset of features between $[\texttt{min\_feats}, \texttt{max\_feats}]$. Data points are grouped into 1–3 clusters by generating cluster centers sampled from a Gaussian distribution. Each datapoint $x \in \mathbb{R}^d$ is assigned a cluster label via its closest cluster center according to Euclidean distance. Then each cluster is assigned its own subset of informative features. Features not selected are masked out and do not influence the label.

2. **Random BNN weights.** A two-layer feedforward neural network with hidden dimension $H = 8$ and $\tanh$ nonlinearity is constructed. Weights and biases are drawn from Gaussian distributions and scaled by random importance weights and scale factors sampled uniformly from given ranges. The masked input features are passed through the random network, producing output logits.

3. **Task-specific adjustments.** Logits are rescaled by a random temperature parameter sampled from a uniform range. For binary tasks, a random bias shift is applied to match a target label prevalence $p \in [0.05, 0.95]$.

4. **Label generation.** The final logits are passed through a sigmoid to produce probabilities, from which labels $Y$ are sampled as Bernoulli (for binary classification) or categorical (for multi-class) random variables.

This procedure defines a flexible family of tasks where both the informative feature subsets and the underlying labeling functions vary across tasks, simulating heterogeneity in feature importance for AFA.

### A.6.3 ADDITIONAL TRAINING DETAILS

At each training step, we sample sequences of length $N$ from the pretraining pool. Feature values are normalized using the mean and variance of each feature within the task sequence. Missingness is introduced either by randomly dropping features (MCAR) or by sampling feature-specific missingness mechanisms from the BNN prior (MAR) that depend only on the baseline covariates $X_0$. The missingness rates vary by feature, and we set the maximum probability of missingness $p(R_j = 0|X_0) \leq 0.5$. A summary of the experimental design is provided in Table 2.

### A.6.4 COMPUTE DETAILS

All experiments were run on a server with 4 NVIDIA H100 NVL GPUs, 2 Intel(R) Xeon(R) Platinum 8480+ CPUs (56 cores each) with 2Tb of memory.

### A.6.5 RUNTIMES

The pretraining procedure for both the predictor and policy using the GP prior (sequence length of 500, 10 features, 100000 training steps) takes approximately 10 hours total on a single GPU.

### A.6.6 HYPERPARAMETERS

For our **L2M** model, we use the same hyperparameter configurations for all our experiments as shown in 3. For pretraining the predictor, all models are trained for 100000 steps with a batch size of 8 tasks (with the exception of **MNIST**, which was trained for 50000 steps). The predictor is trained with the Adam optimizer with the default optimizer parameters and with linear decay. We checkpoint the model at every 500 steps and save the model with the best validation loss.

For the policy, we use a fixed temperature of 0.1 and a batch size of 8 for a total of 50000 training steps, with no learning rate decay. The transformer backbone and predictor weights are also jointly updated, with a lower learning rate of $1 \times 10^{-5}$.

| Hyperparameter | Value |
|---|---|
| Hidden Layer Size | 512 |
| Model Dimension | 256 |
| Number of Layers | 6 |
| Attention Heads | 4 |
| Embedding Depth | 4 |
| Dropout | 0 |
| Predictor Learning Rate | $1 \times 10^{-4}$ |
| Policy Learning Rate | $1 \times 10^{-4}$ |
| Warmup steps | 500 |

Table 3: Transformer Model Hyperparameters

### A.6.7 BASELINES

We describe the baselines used in our experiments, noting several modifications made to improve computational tractability when evaluating across a large number of tasks.

**MLP (Random).** For each evaluation task, we train a two-layer multilayer perceptron (MLP) with hidden dimension 128. The model is trained on randomly selected feature subsets to predict the target label, using a batch size of 64 for 300 epochs. At test time, acquisition actions are chosen uniformly at random, and the MLP is used to make predictions. This task-specific MLP serves as the predictor model for the remaining baselines.

**GDFS** (Covert et al., 2023). The policy network is also a two-layer MLP with hidden dimension 128. In contrast to the original paper, which trains the selector policy using Gumbel-softmax with a temperature decay schedule, we train using the straight-through Gumbel-softmax estimator with a fixed temperature of 0.5.

**DIME** (Gadgil et al., 2023). The reward predictor is also a two-layer MLP with hidden dimension 128. We train the reward predictor using random acquisitions, rather than the $\epsilon$-greedy acquisition strategy with decay as described in the original paper.

**DQN** (Janisch et al., 2019) We adopt the Q-learning framework, where the action-value function $Q(s_t, a)$ estimates the expected return from state $s_t$ after taking action $a$. The optimal acquisition actions are selected by taking the action with the largest Q-value estimate $Q(s_t, a)$.

We consider a dueling network architecture (Wang et al., 2016). The dueling network consists of two MLPs with two hidden layers of dimension 128: one head outputs $V(s_t)$, and the other outputs

$A(s_t, a)$ for all actions $a$. The final Q-value estimate is computed as

$$Q(s_t, a) = V(s_t) + \left( A(s_t, a) - \tfrac{1}{|\mathcal{A}|} \sum_{a'} A(s_t, a') \right),$$

Because the final log likelihood is identical across complete trajectories, we apply a strong discount factor to prioritize early acquisitions that reduce predictive loss. The one-step temporal-difference (TD) target is defined as

$$y_t = r_t + \gamma \max_{a'} Q_{\theta^-}(s_{t+1}, a'),$$

where $r_t$ is the immediate reward, $Q_{\theta^-}$ denotes the target Q-network, and $\gamma = 0.9$ is the discount factor. We train for 200 episodes using an experience buffer of size 10,000, with samples collected via an $\epsilon$-greedy strategy. Training updates use mini-batches of 128 samples, and the target Q-network is synchronized every 4 episodes.

### A.6.8    SOURCE CODE

https://anonymous.4open.science/r/Learning-To-Measure-5635

## A.7 ADDITIONAL RESULTS

### A.7.1 UNCERTAINTY QUANTIFICATION

We perform analogous evaluations as in Figure 3 for the classification tasks. We first evaluate the ability for **L2M** to recover the ground true probabilities in a set of evaluation tasks randomly sampled from the same semi-synthetic BNN task prior used during training. To evaluate uncertainty quantification for classification, we compute the KL divergence and brier score (MSE) between the predicted probabilities and ground truth probabilities. We also additionally show the AUROC to assess the ability for the model to rank samples.

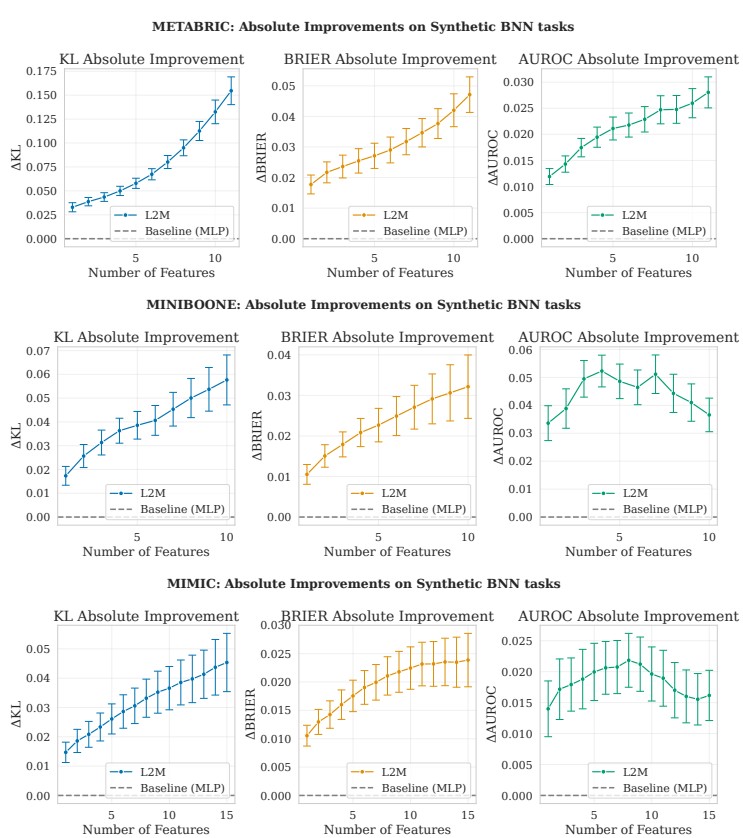

Figure 7: We identify a similar pattern where the quality of uncertainty quantification is better than the task-specific MLP, and the gains are larger as we acquire more features.

Next, we evaluate **L2M** on classification using semi-synthetic tasks built from real labels that were unseen during training. Because these tasks provide hard (binary) labels rather than ground-truth probabilities, we report negative log-likelihood (binary cross-entropy) and Brier score to assess uncertainty. Accordingly, we do not use KL divergence in this setting.

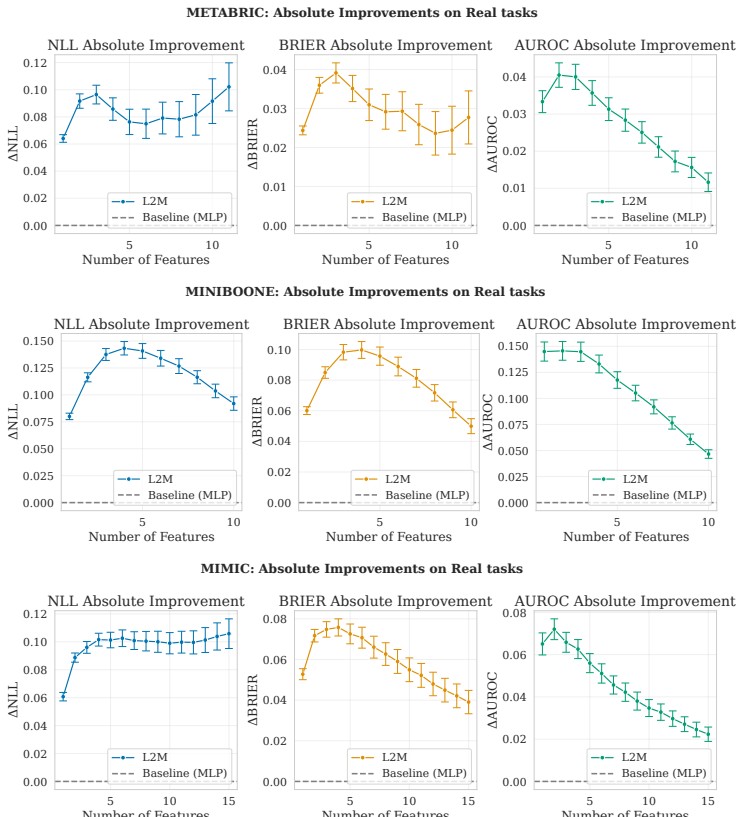

Figure 8: The performance on real tasks is mixed, and dependent on various factors such as whether the pretraining BNN tasks are closely aligned to the real unseen tasks.

### A.7.2 AFA POLICY EVALUATION

**Performance**   We provide additional metrics for quantifying AFA performance.

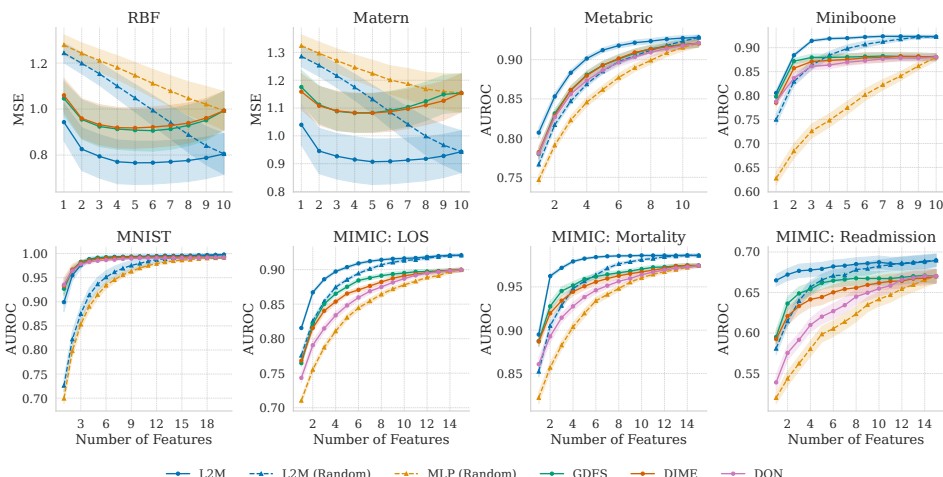

Figure 9: Using the same evaluation tasks and samples, we plot the MSE for the regression tasks and AUROC for the classification tasks

**Policy Visualization**   We demonstrate how our sequence modeling approach is able to learn task-specific policies. We visualize the selected actions by the greedy policy in example evaluation tasks.

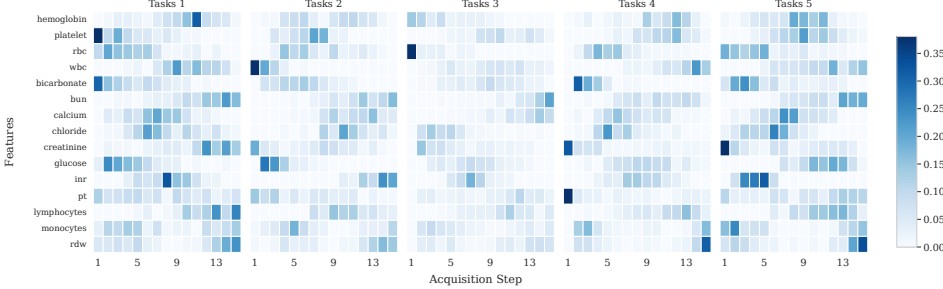

Figure 10: **MIMIC-IV** Dataset: Example feature acquisition for a set of semi-synthetic evaluation tasks constructed using the BNN prior. Each task contains 500 query samples.

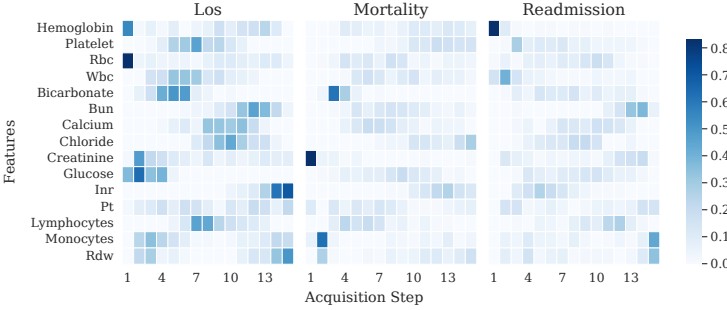

Figure 11: **MIMIC-IV** Dataset: Example feature acquisition for a set of 500 query samples on the semi-synthetic tasks with real labels.

**Acquisition on simulated tasks** We evaluate our model using tasks sampled from our synthetic pretraining prior where underlying feature informativeness is known apriori. We plot the precision and recall achieved by the acquisition method at each budget $k$, as well as the log loss metrics. We use the **MIMIC-IV** test distribution for the covariates $X$, and sample synthetic labels $Y$ using the synthetic prior. We sample 50 test tasks and evaluate with different context lengths.

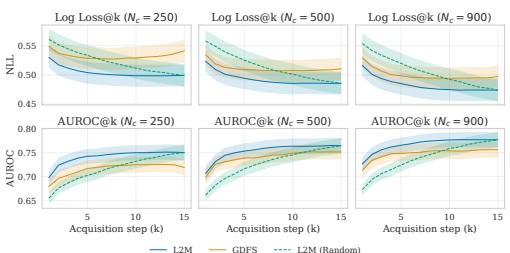 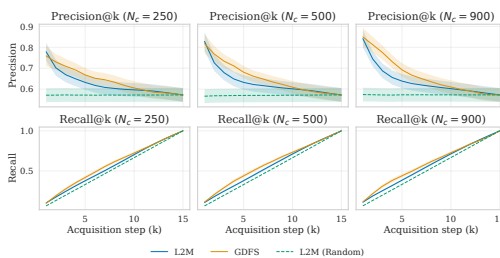

Log Loss and AUROC             Precision and Recall

Figure 12: While the meta-learning approach demonstrates stronger uncertainty quantification (log loss) and discrimination (AUROC) performance at each decision budget, the task-specific approach is slightly more precise when identifying informative features. The task-specific model may be learning a sharper ranking over the features, while the meta-learning approach maintains uncertainty over the optimal acquisition actions.

**Fully synthetic variant** We also train a fully synthetic variant of **L2M**, where each task is defined by covariates sampled from multivariate Gaussian distributions with randomly sampled means and covariances. Across tasks, we allow the feature dimensionality to vary. For evaluation, we sample only simulated tasks with at least 10 features available for acquisition, while leaving the total number of features per task flexible. We note that this setting is particularly challenging for meta-learning, as the task prior must encompass heterogeneous feature distributions with varying dimensionalities. We sample 50 test tasks and evaluate with different context lengths.

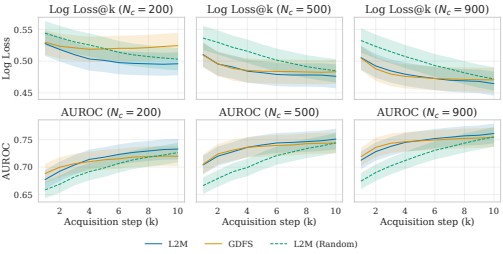 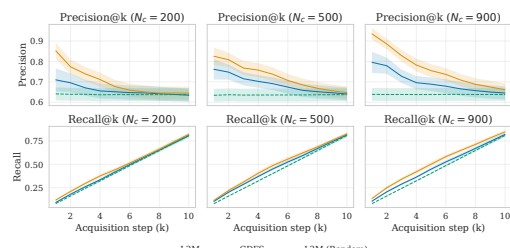

Log Loss and AUROC             Precision and Recall

Figure 13: **L2M** is able to match the performance of the task-specific baseline even in this challenging scenario. However, we find similarly that the task-specific approach is slightly more precise when identifying informative features.

We also evaluate the fully synthetic **L2M** model on real-world tasks. This is a particularly challenging setup, since the model never observes the test task's covariate distribution during pretraining. We denote this baseline as **L2M-Zero**, as the model is applied in a zero-shot fashion to an out-of-distribution setting. We find that **L2M-Zero** acquires features that outperform random selection on the real-world tasks, but its performance degrades substantially compared to an **L2M** model trained directly on the real-world covariate distributions.

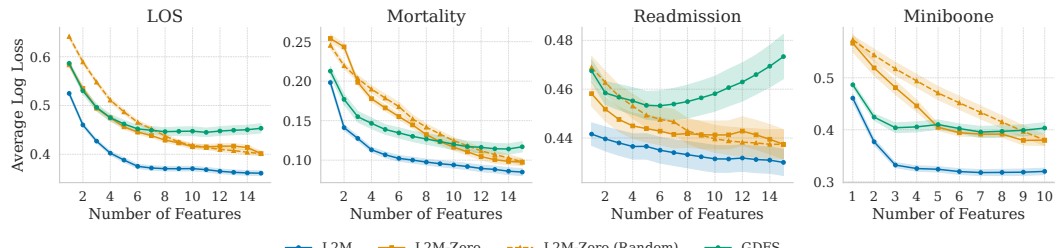

Figure 14: The fully synthetic variant is less effective on real-world tasks, largely because the co-variate distribution it was trained on differs substantially from the real-world data.

**Benefit of Adaptivity**   Tasks such as LOS and mortality prediction are heterogeneous and may depend on different mechanisms across patients. We provide an ablation where we use the same **L2M** model but instead of the per-instance optimal action predicted by the model, we take the most frequently selected action across the entire test set at each step (majority vote). We find that the instance-wise adaptive feature selection is beneficial for some tasks, but other simpler tasks do not require granular acquisition.

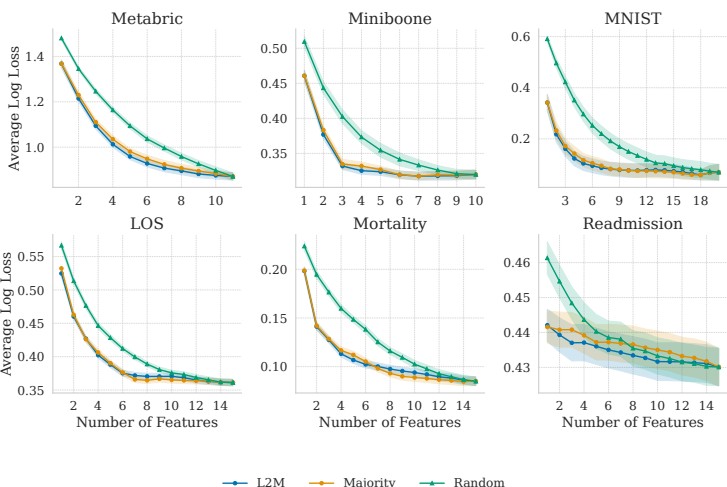

Figure 15: Simpler tasks such as Miniboone do not benefit from instance-wise adaptive selection. We also note that it is difficult to demonstrate the benefit of per-instance adaptivity in small-scale MIMIC experiments.

**Real missingness patterns**   We evaluate **L2M** when the context set exhibits real-world missingness patterns not seen during pretraining. These results should be interpreted with caution. The model was pretrained only on complete cases with synthetically injected missingness, so the evaluation introduces a distribution shift in both patient characteristics and missingness mechanisms. In addition, the query patients are complete cases while the context patients may be partially observed, creating a context–query shift. We therefore present these results as a stress test, not a definitive measure of deployment performance.

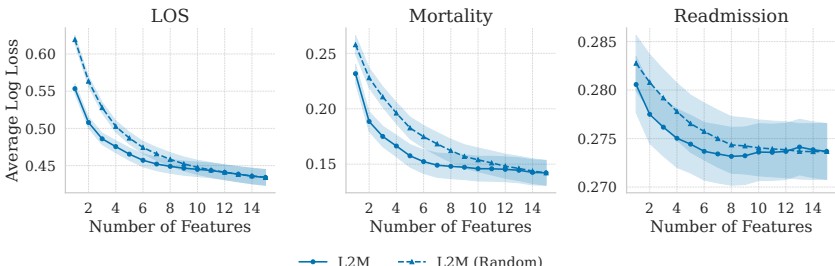

Figure 16: LTM is able to learn policies that improve on random acquisition on unseen natural missingness patterns in real-world data.

## A.8 META-LEARNING ABLATIONS

We provide additional results investigating alternative meta-learning strategies. One common method uses a conditional neural process (CNP) -like (Garnelo et al., 2018) approach to learn compact task representations of variable length context, and use these embeddings for downstream decision-making (Rakelly et al., 2019; Wang et al., 2024). To investigate whether this is an effective approach, we use our same transformer architecture, but perform multihead-attention pooling to learn a task-level embedding instead of full context attention. We use the same pretraining priors, and greedy differentiable policy learning. Additionally, we perform MAML-like (Finn et al., 2017) inner gradient updates for the policy and predictor head for each unseen task encountered during inference, based on the learned CNP task representation.

The results in 17 show that a CNP-like approach leads to degradation in the quality and stability of uncertainty estimates. However, when pretraining on a small, fixed task family (**MNIST**) the task embedding leads to slightly improved performance and data efficiency during earlier acquisition steps. We hypothesize that this is due to the compact representation acting as an information bottleneck and regularizing the model, simplifying greedy policy learning.

We find that gradient-based adaptation at inference offers only modest improvements and rapidly overfits, indicating that the compact task representation acts as a bottleneck that the test-time adaptation cannot overcome. The main exception is the **MIMIC-IV** mortality task, where we see consistent gains. We hypothesize this is because very low positive prevalence is underrepresented in the pretraining prior, so adaptation corrects this mismatch.

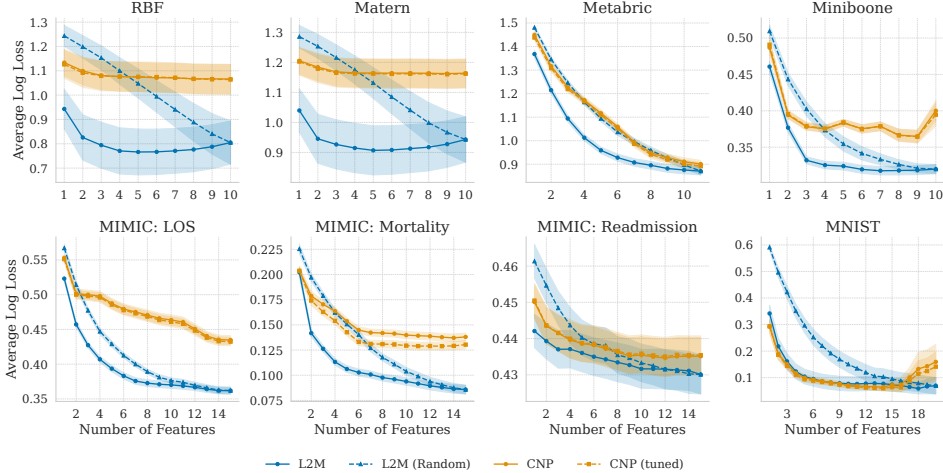

Figure 17: A CNP-like approach that learns a compact task representation via attention pooling leads to less stable uncertainty estimates, especially when pretrained on a diverse task prior.

## A.9   LLM USAGE

LLMs were used to assist in code generation, specifically computing evaluation metrics and code for generating result figures. All AI-assisted code was checked for accuracy. LLMs were also used for checking grammar and formatting assistance.

