# OpenReview forum: "Learning-To-Measure: In-Context Active Feature Acquisition"
_ICLR.cc/2026/Conference — Submitted to ICLR 2026_

### Official Review · Reviewer_oT93 · 2025-10-31

**Soundness:** 2
**Presentation:** 3
**Contribution:** 2
**Rating:** 4
**Confidence:** 3

**Summary:**

The paper introduces and studies meta–active feature acquisition: from retrospective datasets where features are missing, learn a single meta, in-context policy that sequentially acquires features without per-task retraining. The method, Learning-to-Measure (L2M), uses a transformer to (i) estimate output uncertainty for partially observed inputs and (ii) choose the next feature by minimizing a surrogate for greedy conditional mutual information (CMI). It also provides a causal result stating that, under missingness-at-random, exclusion, and positivity, the target CMI on retrospective datasets equals the CMI computed from complete cases. It empirically tests L2M on synthetic GP and semi-synthetic (Metabric, MiniBooNE, MIMIC-IV) tasks, as well as an MNIST block-acquisition task.

**Strengths:**

- **Theoretical soundness.** The paper provides a novel and clear meta-formulation of Active Feature Acquisition and theoretically extends single-task greedy surrogates to actions conditioned on variable-length contexts.
- **Flexible implementation.** The authors first pre-train a meta-predictor, then train a meta (blocked) policy that makes training more seamless on retrospective datasets with missing features.
- The paper provides **reasonable empirical evidence** that L2M helps when labels are scarce or when the missingness is significant. The paper is also transparent about assumptions and limitations.
- The method and proofs are clearly written, as well as the architecture and training details for reproducibility.

**Weaknesses:**

- **Baselines** are simple MLP or RL AFA methods trained per task. Meta-learning methods with potential minor per-task tuning are missing.
    - Some tasks show L2M close to “static” or even random strategies, which suggests limited headroom and potentially lower performance to stronger baselines. On this line, authors could add an oracle to bound possible gains and contextualize the significance of their method.

I don’t find any other major actionable weaknesses; my remaining primary concern is the significance of the contributions, since the pieces make only minor adaptations to existing theory or ideas. This prevents me from increasing my score, but I am open to changing my opinion regarding the novelty based on the author’s response.

### Minor weaknesses

None of these items has a significant impact on my score, but I encourage the authors to consider incorporating some of them.

1. **Architecture choices are not substantiated.** They make some ad-hoc choices (e.g.,  masked features concatenated with masks). However, no ablations are showing whether their different design choices that depart from standard practice matter.
2. **Causal identifiability is brittle.** The paper acknowledges that MAR, exclusion, and positivity are strong assumptions. Still, given that it uses synthetic data, I would encourage an empirical stress test (e.g., simulate MNAR or partial violations of positivity) to assess robustness to mild violations of these assumptions.
3. **Pretraining priors are small.** I would encourage exploring an experiment that pre-trains on fully synthetic Bayesian networks in tabPFN-style [1]. This is a natural extension that could significantly increase the empirical significance.
4. The method section says the framework can leverage pretrained LLMs, but experiments do not evaluate LLM-based backbones, and effectiveness is unclear. This should be stated as future work or demonstrated empirically.
5. **Cost-sensitive variants.** Realistic AFA requires per-feature costs; even a small study (variable costs on MIMIC-IV labs) would strengthen the paper.
6. **L121**: Typo “Time-invaryiant”

[1] Hollmann, N., Müller, S., Eggensperger, K., & Hutter, F. (2022). TabPFN: A Transformer That Solves Small Tabular Classification Problems in a Second. *ArXiv*. https://arxiv.org/abs/2207.01848

**Questions:**

No specific questions; please take a look at the weaknesses above.

---

> ### Author Response · Authors · 2025-11-22
>
> Thank you for a thoughtful and constructive review. We address your concerns in the following.
>
> **[W1] Meta-learning baselines**
> > Baselines are simple MLP or RL AFA methods trained per task. Meta-learning methods with potential minor per-task tuning are missing.
>
> Thank you for raising the point regarding baselines. To our knowledge, no prior work provides meta-learning baselines for AFA. To address your concern, we create additional meta-learning baselines using a conditional neural process (CNP) to learn task embeddings of variable-length contexts. Learning compact task embeddings without explicit task labels is a common strategy in the meta-RL literature [1,2]; here we evaluate its applicability to our settings. We then add a CNP + MAML [3] -like baseline that takes inner gradient steps on the inference tasks to adapt the predictor and policy head. While simple, these would constitute baselines adapting meta-learning to AFA with per-task finetuning.
>
> We include the new results in Appendix A.8 (figure 15), showing that CNP-style pooling yields less reliable uncertainty and becomes unstable, with only modest early-step gains on simpler, low-diversity tasks. We discuss how a global pooled embedding acts as an information bottleneck and note when it helps vs. hurts.
>
> > [1] Rakelly, K., Zhou, A., Finn, C., Levine, S., & Quillen, D. (2019, May). Efficient off-policy meta-reinforcement learning via probabilistic context variables. In International conference on machine learning (pp. 5331-5340). PMLR.
>
> > [2] Wang, Z., Zhang, L., Wu, W., Zhu, Y., Zhao, D., & Chen, C. (2024). Meta-DT: Offline meta-RL as conditional sequence modeling with world model disentanglement. Advances in Neural Information Processing Systems, 37, 44845-44870.
>
> > [3] Finn, C., Abbeel, P., & Levine, S. (2017, July). Model-agnostic meta-learning for fast adaptation of deep networks. In International conference on machine learning (pp. 1126-1135). PMLR.
>
> **[W2] Empirical significance**
> > Some tasks show L2M close to “static” or even random strategies, which suggests limited headroom and potentially lower performance to stronger baselines. On this line, authors could add an oracle to bound possible gains and contextualize the significance of their method.
>
> We agree with the reviewer’s assessment that close to random performance on MIMIC tasks shows limited headroom. It is difficult to showcase the benefit of adaptivity (per-instance differential acquisition) on such noisy small-scale tasks (such as readmission/length of stay prediction), compared to selecting the population-level optimal acquisitions at each step. To improve empirical significance, we updated the MIMIC experiments in the manuscript (figure 4,5, 7-15) with an increased number of both baseline conditioning and acquisition features.
>
> While we cannot construct an “oracle” for instance-wise adaptive selection in the sense that it upper bounds performance in real data, we demonstrate the correctness of our method using synthetic tasks with known feature informativeness compared to such an “oracle”. We evaluate via recall and precision at each budget constraint (Figure 12).

---

> > ### Author Response · Authors · 2025-11-22
> >
> > **Novelty and Significance**
> > > I don’t find any other major actionable weaknesses; my remaining primary concern is the significance of the contributions, since the pieces make only minor adaptations to existing theory or ideas. This prevents me from increasing my score, but I am open to changing my opinion regarding the novelty based on the author’s response.
> >
> > We believe that in the context of AFA literature, our contributions are significant.  To our knowledge, this is the first work that studies the application of meta-learning to the AFA problem, providing the first empirical evidence that amortized policy learning is possible for AFA using our autoregressive sequence loss. Second, we incorporate an objective aligned with practical considerations of offline learning, i.e., handling retrospective missingness. Our resulting sequence modeling framework effectively propagates uncertainty across varying context lengths and retrospective feature missingness, contributing to the robustness of our AFA method. We provide novel empirical evidence supporting these claims.
> >
> > Our contributions are both conceptual and methodological, expanding AFA to the foundation model paradigm, a capability missing and unreliable in the current paradigm. We strongly believe that while each piece of the solution leverages existing ideas, the cumulative contribution will open the path to further methodological developments in this direction, subsequently bridging in-context information-seeking and AFA.
> >
> > **Input representation**
> > > Architecture choices are not substantiated. They make some ad-hoc choices (e.g., masked features concatenated with masks). However, no ablations are showing whether their different design choices that depart from standard practice matter.
> >
> > We note that the concatenation of missingness indicators as a mask in our architecture is not our novel contribution and is standard in past AFA works. The input representation of the observed feature set is flexible (e.g., a set encoder approach, tokens in natural language, or masked features + mask). We have added this point to line 321 the manuscript for clarity.
> >
> > **Sensitivity to violations**
> > > Causal identifiability is brittle. The paper acknowledges that MAR, exclusion, and positivity are strong assumptions. Still, given that it uses synthetic data, I would encourage an empirical stress test (e.g., simulate MNAR or partial violations of positivity) to assess robustness to mild violations of these assumptions.
> >
> > Thank you for your comment and we acknowledge the limitations of these assumptions, but prefer to be transparent about it. While we leave a rigorous study of MNAR and positivity to future work, we stress test our method using real-world missingness patterns from our MIMIC experiments with unknown missingness mechanisms (figure 13). We also provide additional simulations for violations (We will update the manuscript with these results as soon as this is ready.)
> >
> > **Fully synthetic baseline**
> > > Pretraining priors are small. I would encourage exploring an experiment that pre-trains on fully synthetic Bayesian networks in tabPFN-style [1]. This is a natural extension that could significantly increase the empirical significance.
> >
> > To address your concern, we provide an additional baseline where L2M is pretrained on fully synthetic data of variable feature dimension, showing that the policy network can be pretrained across a flexible range of covariate distributions. (We will update the manuscript with these results as soon as this is ready.)
> >
> > **Other comments**
> > > The method section says the framework can leverage pretrained LLMs, but experiments do not evaluate LLM-based backbones, and effectiveness is unclear. This should be stated as future work or demonstrated empirically.
> >
> > > Cost-sensitive variants. Realistic AFA requires per-feature costs; even a small study (variable costs on MIMIC-IV labs) would strengthen the paper.
> >
> > We are actively pursuing both AFA with LLMs and non-myopic cost-sensitive planning but require additional methodological contributions. Variable per-feature costs in greedy acquisition are handled via a confidence threshold or budget, which yields the same myopic policy up to a rescaling. We will acknowledge these as limitations (line 483-5), stating that a rigorous study of LLM-based acquisition and non-myopic cost-sensitive planning is future work.
> >
> > We thank you for your time and consideration. We have incorporated your suggestions (in red) and believe the revisions strengthen the empirical evidence and clarify our contributions. If you have additional questions, please let us know. In light of our response, we hope you consider improving the score.

---

### Official Review · Reviewer_F7yg · 2025-10-31

**Soundness:** 3
**Presentation:** 3
**Contribution:** 3
**Rating:** 4
**Confidence:** 3

**Summary:**

This paper addresses the problem of active feature acquisition (AFA) by reformulating it as a meta-learning task. The authors argue that traditional, single-task AFA methods are not scalable and struggle with retrospective datasets that have systematic missingness and scarce labels. To solve this, the paper introduces Learning-to-Measure (L2M), an in-context learning framework. L2M uses a sequence model (a Transformer) that can adapt to a new, unseen task by conditioning on a "context" of a few labeled samples from that task. Experiments on synthetic and semi-synthetic tasks derived from real-world tabular datasets (MIMIC-IV, Metabric, etc.) show that L2M outperforms task-specific AFA baselines (like GDFS and DIME), especially in the challenging (and realistic) regimes of high missingness and scarce labeled data (small context sizes).

**Strengths:**

-	While the practical usefulness is somewhat questionable, framing AFA as a meta-learning or in-context learning problem is a departure from the previous works, which directly addresses the critical issue of scalability and adaptation that plagues traditional single-task AFA models.

**Weaknesses:**

-	The motivation for a non-greedy AFA framework is sound—identifying when certain features act as strong indicators for acquiring subsequent, highly informative ones (as illustrated in the paper’s chest-pain triage example) is indeed valuable. However, similar motivations have been explored in recent work [A], which also highlights the limitations of conditional mutual information objectives. Moreover, reinforcement learning–based approaches that incorporate discounted future rewards are already capable of addressing such sequential dependency issues, weakening the novelty claim of the proposed formulation. Moreover, the proposed method adopts a greedy, one-step CMI maximization policy. This is a significant drawback, as it fails to capture multi-step acquisition strategies and may result in myopic decisions, as highlighted in [A].
-	It is unclear how Equation (1) leads to different acquisition behaviors when missingness indicators are included as input features. The experiments do not explicitly analyze or visualize how the model leverages these indicators to guide acquisition decisions.
-	The paper claims that the feature acquisition policy is meta-learned across datasets. However, in practice, the policy generates a categorical distribution over a fixed feature dimension $d$, assuming a shared feature space across all tasks. This assumption substantially limits the method’s applicability to real-world settings where feature spaces often differ across domains or datasets.
-	The missingness patterns in experiments are synthetically constructed by the authors rather than drawn from naturally incomplete real-world datasets. Evaluating the approach on datasets with naturally occurring missingness would provide stronger evidence of its robustness and practical utility.

[A] Norcliffe et al., “Stochastic Encodings for Active Feature Acquisition,” ICML 2025.

**Questions:**

- Have the authors considered ablations in settings known to be difficult for greedy policies (e.g., high feature redundancy)? How would L2M compare to a non-myopic, multi-step lookahead policy such as [A], even a simple one, built on top of L2M's excellent uncertainty-quantifying sequence model?

- The procedure for setting the blocked policy is not clearly described. How are the blocked features determined (i.e., how is R_j=0assigned)?

- The meta-learning is demonstrated on tasks sampled from synthetic (GP) or semi-synthetic (BNN) priors. It is unclear how well this pre-training will generalize to a distribution of truly diverse real-world tasks. The mixed results on "real tasks" (Figure 8) suggest the BNN prior may not be rich enough, which could undermine the central meta-learning claim.


[A] Norcliffe et al., “Stochastic Encodings for Active Feature Acquisition,” ICML 2025.

---

> ### Author Response · Authors · 2025-11-22
>
> We thank the reviewer for the helpful and detailed comments. We hope that our clarifications will address your concerns and encourage you to reassess our work.
>
> **[W1] Greedy acquisition**
> > Similar motivations have been explored in recent work [A], which also highlights the limitations of conditional mutual information objectives. Moreover, reinforcement learning–based approaches that incorporate discounted future rewards are already capable of addressing such sequential dependency issues, weakening the novelty claim of the proposed formulation. Moreover, the proposed method adopts a greedy, one-step CMI maximization policy. This is a significant drawback, as it fails to capture multi-step acquisition strategies and may result in myopic decisions, as highlighted in [A].
>
> We appreciate the concern and provide some clarifications:
>
> We acknowledge that RL-based methods that formulate the AFA problem as an MDP is indeed theoretically optimal, as it maximizes cumulative reward via long-term/multi-step planning instead of greedy myopic reward. Greedy methods are used as a practical approximation, and have been shown to perform reliably (and often outperform) compared to RL-based methods that suffer from practical training difficulties despite being theoretically appealing [1]. We provide a DQN baseline showing inferior performance to greedy methods, showing the difficulties fitting task-specific RL methods on different dataset sizes, label and missingness mechanisms without extensive tuning. While RL methods work well in other domains, we believe the difficulty of RL-based methods in AFA arises from trying to approximate value functions offline using limited historical interaction data, and that the number of possible trajectories grows exponentially as the number of features increases, making credit assignment (in realistic settings with small loss changes per acquisitions) difficult. This makes greedy methods practically appealing.
>
> Alternatively, methods like [A] (while it does not explicitly perform multistep RL planning) improve on greedy CMI acquisition by approximating a different CMI objective that does not marginalize out unacquired features. However, we note that their method requires a specific task-specific latent variable architecture with feature-specific encoders, and samples the latent variable multiple times at each acquisition step. These design choices make it non-trivial to adapt to our meta-learning formalism in a scalable manner.
>
> We emphasize that our main conceptual contribution, which departs from prior work, is the missingness-aware meta-learning formalization of the AFA problem, which we believe is better aligned to practice. We then provide a proof of concept showing that greedy AFA policies can be meta-learned with sequence models, instilling robustness and leading to concrete gains compared to task-specific policies. We emphasize that the meta-learning framework enables reliable propagation of uncertainty from varying context sizes and varying retrospective missingness. That being said, we will acknowledge greedy decision-making as a limitation and expand our discussion (Appendix A.4).
>
> > [1] Schütz, V., Wu, H., Rezvan, R., Aronsson, L., & Chehreghani, M. H. (2025). AFABench: A Generic Framework for Benchmarking Active Feature Acquisition. arXiv preprint arXiv:2508.14734.
>
> **[W2] Missingness Indicators**
>
> > It is unclear how Equation (1) leads to different acquisition behaviors when missingness indicators are included as input features. The experiments do not explicitly analyze or visualize how the model leverages these indicators to guide acquisition decisions.
>
> Conceptually, equation (1) should be considered an ‘ideal’ objective function when historical data does not have missingness. Naturally occurring data often has missing features, and a reliable AFA agent must capture and propagate uncertainty from missingness patterns in historical data in order to estimate the information gain of acquiring a feature reliably. The main change in acquisition behavior is multifold. First, the pattern of missingness must be MAR to obtain an unbiased estimate of Equation (1) from the data with missingness (Theorem 3.4).
>
> Second, during training the model leverages these missingness indicators implicitly in obtaining a better estimate of posterior uncertainty, and learning over trajectories where support over data is available. We hope this clarifies how missingness indicators are leveraged by the framework during training. Further, see our response related to the blocked policy.
>
> Practically, feature sets are represented as masked features with concatenated missingness indicators represented as masks. This concatenation of missingness indicators is standard in prior AFA work and not a contribution of our method; accordingly, we do not add experiments specific to this component. We have clarified this in the manuscript (lines 320–321).

---

> > ### Author Response · Authors · 2025-11-22
> >
> > **[W3] Policy dimensionality**
> > > The paper claims that the feature acquisition policy is meta-learned across datasets. However, in practice, the policy generates a categorical distribution over a fixed feature dimension , assuming a shared feature space across all tasks. This assumption substantially limits the method’s applicability to real-world settings where feature spaces often differ across domains or datasets.
> >
> > We note that our method is in fact flexible to the feature dimensionality of the input datasets, letting the policy output a categorical distribution over different sets of features via masking. To substantiate the practicality of our method, we introduce an additional baseline where the L2M model is pretrained completely on synthetic data (both X and Y) of varying feature spaces/dimensions and applied downstream in a “zero-shot” manner to our real-world evaluation tasks  (We will update the manuscript with these results as soon as this is ready.)
> >
> > **[W4] Real-world missingness**
> >
> > > The missingness patterns in experiments are synthetically constructed by the authors rather than drawn from naturally incomplete real-world datasets. Evaluating the approach on datasets with naturally occurring missingness would provide stronger evidence of its robustness and practical utility.
> >
> > Thank you for the suggestion. We include a new evaluation using real missingness patterns in MIMIC (figure 14), using the same tasks. For testing our method, we cannot actually evaluate the acquisition for features where we do not have the ground truth value retrospectively available, hence our choice of evaluations.
> >
> > **[Q1] Non-myopic setting**
> > > Have the authors considered ablations in settings known to be difficult for greedy policies (e.g., high feature redundancy)? How would L2M compare to a non-myopic, multi-step lookahead policy such as [A], even a simple one, built on top of L2M's excellent uncertainty-quantifying sequence model?
> >
> > While we do not have empirical results showing when greedy methods fail, we will provide additional discussion (Appendix A.4) using the concept of adaptive submodularity [2], which theoretically characterizes when greedy methods fail. Intuitively, greedy acquisition will fail if added predictive value only comes from acquiring some features jointly. In this case, planning methods will foresee the need for joint acquisition as opposed to greedy methods.
> >
> > As previously mentioned, extending methods such as [A] is non-trivial to the meta-learning setting with planning, and we are actively pursuing new methods to learn non-myopic policies with in-context learning for future work. We note that this problem for general RL is also non-trivial, and in-context RL is a new and active area of research [3]. To our knowledge, our method is the first application of this framework to AFA, moving the needle of the state of AFA research.
> >
> > > [2] Golovin, D., & Krause, A. (2011). Adaptive submodularity: Theory and applications in active learning and stochastic optimization. Journal of Artificial Intelligence Research, 42, 427-486.
> >
> > > [3] Moeini, A., Wang, J., Beck, J., Blaser, E., Whiteson, S., Chandra, R., & Zhang, S. (2025). A survey of in-context reinforcement learning. arXiv preprint arXiv:2502.07978.

---

> > > ### Author Response · Authors · 2025-11-22
> > >
> > > **[Q2] Blocking**
> > > > The procedure for setting the blocked policy is not clearly described. How are the blocked features determined (i.e., how is R_j=0 assigned)?
> > >
> > > We interpret this question as how blocking is implemented in practice during the pretraining procedure. At each training step, an in-context dataset generated from a synthetic task is sampled. $R_j = 0$ is determined from the sampled synthetic missingness mechanism for the task. This enables the sequence model to encounter diverse missingness mechanisms during pretraining. During policy training, we set $R_j = 0$ for features not observed in each training task. Now, actions with $R_j=0$ are blocked, that is, the policy never acquires those features, since we cannot assess the utility of acquiring that feature without its ground-truth observation or a full generative model. We make a clarifying statement on line 328. If we missed something, please let us know and we’re happy to clarify further.
> > >
> > > **[Q3] Synthetic priors**
> > > > The meta-learning is demonstrated on tasks sampled from synthetic (GP) or semi-synthetic (BNN) priors. It is unclear how well this pre-training will generalize to a distribution of truly diverse real-world tasks. The mixed results on "real tasks" (Figure 8) suggest the BNN prior may not be rich enough, which could undermine the central meta-learning claim.
> > >
> > > The idea of pre-training with synthetic priors is not new, and models like TabPFN [4] provide an existence proof that more complex synthetic prior designs and increasing scale can achieve good performance on diverse real-world datasets with flexible dimensionality of rows and columns. Our proof-of-concept leverages this successful example to pretrain models for AFA, demonstrating promising initial results. Figure 8 shows that such meta-learning still outperforms task-specific approaches (since the delta is positive), but the pattern of larger gains as more features are acquired does not necessarily hold since performance may saturate at earlier acquisition steps in real tasks. As suggested in the discussion, we note that prior specification research and evaluation on diverse real-world tasks is an important aspect of future work (see Line 482)
> > >
> > > > [4] Grinsztajn, L., Flöge, K., Key, O., Birkel, F., Jund, P., Roof, B., ... & Hutter, F. (2025). TabPFN-2.5: Advancing the State of the Art in Tabular Foundation Models. arXiv preprint arXiv:2511.08667.
> > >
> > > We appreciate your time and consideration. We have updated the manuscript to incorporate your suggestions (in blue and red). If you have additional questions, please let us know. In light of our response, we hope you consider improving the score.

---

### Official Review · Reviewer_zLhx · 2025-11-01

**Soundness:** 3
**Presentation:** 3
**Contribution:** 3
**Rating:** 6
**Confidence:** 2

**Summary:**

This work extends the existing active feature acquisition (AFA) from single-task to multi-task scenarios based on the meta-learning paradigm — Learning-to-Measure (L2M). Specifically, the authors highlight that maximizing the conditional mutual information (CMI) between adding the new features $X_j$ and the prediction $Y$ based on the current state $\underline{X}_t$ should consider proper uncertainty quantification for the unseen task under missingness in historical data. After proposing the theoretical proof of the identification of CMI with missingness in the current state, they propose a surrogate optimization problem to estimate uncertainty quantification during the pre-trained (meta-learning) stage and train the feature acquisition policy model and the original models. The empirical results demonstrate the competitive uncertainty estimation and query performance on both synthetic and real-world datasets.

**Strengths:**

1. This work provides the solid theoretical proof to claim the feasibility of the estimation of the CMI from the missingness in the historical data.
2. The authors carefully construct the surrogate optimization problem with a clear roadmap from the theoretical proof.
3. The empirical results show that the feasibility of their ideas could be used for the transformers model and real-world datasets, which would appeal to more researchers to follow up on this direction.

**Weaknesses:**

1. While the authors provide examples about the missing features on the historical data, such as clinical data in Sec 1, they assume that the features are time-invariant in Sec 3, which sounds conflicting to me. Is my understanding of the time-invariant features incorrect?
2. Although the authors provide good demonstrations on the real-world tabular datasets, these datasets' dimensions still seem not big enough. I encourage the authors could try large-dimensional tabular datasets such as (1) Musk, (2) Bioresponse, and (3) Diabetes 130-US Hospitals.
3. Typo in Line 242, *with sufficient flexibility ~~while~~ while providing principled...*.

- (1) Chapman, D. & Jain, A. (1994). Musk (Version 1) [Dataset]. UCI Machine Learning Repository. https://doi.org/10.24432/C5ZK5B.
- (2) https://www.openml.org/search?type=data&status=any&id=4134&sort=runs
- (3) Clore, J., Cios, K., DeShazo, J., & Strack, B. (2014). Diabetes 130-US Hospitals for Years 1999-2008 [Dataset]. UCI Machine Learning Repository. https://doi.org/10.24432/C5230J.

**Questions:**

1. In your illustrations in Figure 2, the missingness of a column might only be for a few records (rows). During training, how do you handle missing records and those that remain?

---

> ### Author Response · Authors · 2025-11-22
>
> We thank the reviewer for the helpful comments and for pointing out the typos. We address the concerns below:
>
> **[W1] Time-invariant feature assumption**
> > While the authors provide examples about the missing features on the historical data, such as clinical data in Sec 1, they assume that the features are time-invariant in Sec 3, which sounds conflicting to me. Is my understanding of the time-invariant features incorrect?
>
> We make a simplifying assumption about the features as time-invariant, meaning the underlying values of the features do not change across “steps” of the acquisition trajectory; acquiring a feature i at a given time step simply reveals the underlying static value $x_i$. This assumption is also applied to missingness in historical data. For example, in the chest pain triage example, acquiring lab tests in the first “step” and acquiring a subsequent chest X-ray in the second step, but interchanging these steps (acquiring a chest X-ray first, then labs) does not change the underlying values. We acknowledge that this assumption may not hold for longer-term acquisitions (across hospitalization days or over multiple visits), but note that accounting for time-varying features requires a dynamic model.
>
> **[W2] Dataset scale**
> > Although the authors provide good demonstrations on the real-world tabular datasets, these datasets' dimensions still seem not big enough. I encourage the authors could try large-dimensional tabular datasets such as (1) Musk, (2) Bioresponse, and (3) Diabetes 130-US Hospitals.
>
> Our current experiments are a proof-of-concept constrained by compute. However, publicly available models like TabPFN [1] provide a strong existence proof that our framework can be scaled and pretrained on much larger synthetic datasets (50000 rows, 2000 features), and applied downstream to adaptively acquire features in the datasets cited. Our present evaluation focuses on moderately sized inference tasks (1000 rows, 20 features), though extension to larger datasets is an important engineering problem. We extend our discussion to reflect this (line 489).
>
> > [1] Grinsztajn, L., Flöge, K., Key, O., Birkel, F., Jund, P., Roof, B., ... & Hutter, F. (2025). TabPFN-2.5: Advancing the State of the Art in Tabular Foundation Models. arXiv preprint arXiv:2511.08667.
>
> **[Q1] Training Procedure**
> > In your illustrations in Figure 2, the missingness of a column might only be for a few records (rows). During training, how do you handle missing records and those that remain?
>
> Thank you for raising this concern. We acknowledge that Figure 2 only provides a very high-level view of the training procedure, and the exact training procedure is deferred to Algorithm 2 in Appendix A.5.2. The training procedure is consistent across all tasks with low or high degrees of missingness. We place an entire task dataset in a contiguous sequence, and all rows are used (so no rows are discarded). R is a mask variable that represents the availability (both retrospectively missing for only a few records or the entire column doesn’t exist for the task), and this mask is concatenated to each row. We make these clarifications in the Figure 2 caption, as well as line 321.
>
> If you have additional questions, please let us know. In light of our response, we hope you consider improving the score.

---

### Author Response · Authors · 2025-11-30

We are sincerely grateful to the reviewers for their detailed and constructive feedback.

We summarize the updates to the manuscript here:

1. **Clarified limitations of greedy acquisition.**
*(Reviewer **F7yg**)* We expanded the discussion of greedy selection via adaptive submodularity and now explicitly describe cases where myopic selection can fail *(Sec. A.4)*.

2. **Added meta-learning baselines.**
*(Reviewer **oT93**)* We introduced a Transformer-CNP baseline for task embeddings and a Transformer-CNP+MAML variant with inner-loop gradient adaptation of both the predictor and policy at inference time. These ablations show that task-embedding–based meta-learning is effective only under relatively narrow, finite task priors. *(Fig. 17, Sec. A.7.3)*.

3. **Diversity of task prior.**  *(Reviewer **F7yg**, **oT93**)* We pretrained our method on fully synthetic data with flexible feature dimensionality and covariate distributions to test robustness to heterogeneous feature spaces. We show that the model can still learn effective acquisition policies in this more challenging setting. *(Figs. 13–14, Sec. A.7.2)*.

4. **Included “oracle” checks.**
*(Reviewer **oT93**)* On synthetic tasks with known informative features, we now report precision/recall@k and feature-recovery curves versus baselines to validate selection quality, showing that our method acquires features that significantly improve over random selection *(Figs. 12-13, Sec A.7.2)*.

5. **Increased real-data experiment scope.** *(Reviewer **oT93**)* Some original experiments offered limited headroom and opportunity to observe meaningful performance differences. We expanded the MIMIC-IV evaluation setup to provide a better assessment of real-world performance.

6. **Real-world missingness.** *(Reviewer **F7yg**)* We added a stress test on MIMIC-IV using naturally occurring EHR missingness that was not seen during training. Despite being trained only on synthetic missingness patterns, our method still learns acquisition policies that outperform random feature acquisition *(Fig 16, Sec A.7.2)*.

These changes address the main concerns regarding baselines, scope and limitations, and empirical validation. We hope the revisions clarify our contributions and strengthen the paper.

---

### Meta-Review · Area_Chair_oApW · 2025-12-28

**Summary:**

The paper introduces and studies meta–active feature acquisition: from retrospective datasets where features are missing, learn a single meta, in-context policy that sequentially acquires features without per-task retraining. Reviewers had raised main comments, such as novelty, method clarity, unclear/missing ablations, limited experiments etc.

**Reviewer Concerns:**

no one engaged in rebuttal so I cannot see any ratings update or not.
The only reviewer who gave rating of 6 is not an expert in the field. Similarly, as AC I am not an expert in this field either.

**Reviewer Scores:**

NA

---

### Decision · Program_Chairs · 2026-01-26

Reject